# Transparent and high-porosity aluminum alkoxide network-forming glasses

Zihui Zhang [1] & Yingbo Zhao [1,2] ✉

Metal-organic network-forming glasses are an emerging type of material capable of combining the modular design and high porosity of metal-organic frameworks and the high processability and optical transparency of glasses. However, a generalizable strategy for achieving both high porosity and high glass-forming ability in modularly designed metal-organic networks has yet to be developed. Herein, we develop a series of aluminum alkoxide glasses and monoliths by linking aluminum-oxo clusters with alcohol linkers. A bulky monodentate alcohol modulator is introduced during synthesis and act as both network plasticizer and pore template, which can be removed by the subsequent solvent exchange to give gas accessible pores. Glasses synthesized with the modulator template exhibit well-defined glass transitions in their as-synthesized form and high surface areas up to 500 $m^2/g$ after activation, making them among the most porous glassy materials. The aluminum alkoxide glasses also have optical transparency and fluorescent properties, and their structures are elucidated by pair-distribution functions, spectroscopic and compositional analysis. These findings could significantly expand the library of microporous metal-organic network-forming glasses and enable their future applications.

Metal-organic network-forming glasses are an emerging type of glassy material that can combine high porosity, optical transparency, chemical robustness, and defect-free interfaces[1,2]. Conceptually, a glassy network with micropores can be created from an oxide glass structure by replacing metal and oxygen atoms with metal clusters and organic linkers, which would result in a coordinative network. However, synthetically preparing such porous glassy networks is fundamentally challenging: obtaining glasses of coordinative networks is challenging due to their high melting point and low thermal stability[1]; Retaining microporosity in glasses is even more challenging because glasses tend to have structures similar to their corresponding supercooled liquids which are usually nonporous[3]. Consequently, although melt-quenching, perturbative and other methods have been developed to vitrify MOFs[4-14], the number of known glassy coordinative networks is still very limited compared to the vast library of crystalline metal-organic frameworks (MOFs), and even smaller fractions of these networks have gas accessible microporosity[4-10]. Thus, it would be valuable

to develop a generalizable strategy for synthesizing coordination network glasses with modular designability, high porosity and glass-forming ability, which could enable the incorporation of functional motifs and achieve other desirable properties such as optical properties alongside porosity and processability.

Bottom-up synthesis represents an important strategy to develop glassy coordinative networks with high porosity. It was discovered that porous titanium phenolate and carboxylate networks can be directly synthesized as monoliths by a solvent evaporation approach[15,16]. However, well-defined glass transitions were not observed for these titanium phenolates and carboxylates with gas accessible porosity, which is associated with a trade-off between porosity and glass forming ability: while linker rigidity is required to create pores, it would also result in high tendency for crystallization and give low glass forming ability. Such trade-off has fundamentally limited the structural diversity, porosity and tunability of bottom-up synthesized metal-organic network-forming glasses. Thus, a new synthetic strategy is required to

[1]School of Physical Science and Technology, ShanghaiTech University, Shanghai, China. [2]Shanghai Key Laboratory of High-Resolution Electron Microscopy, ShanghaiTech University, Shanghai, China. ✉e-mail: zhaoyb2@shanghaitech.edu.cn

mitigate such trade-off and produce modularly designed coordinative networks with well-defined glass transition and high gas accessible porosity.

In this report, we develop a series of microporous coordinative glasses and monoliths made from aluminum-oxo clusters (AlOCs) linked with alcohols linkers, which are synthesized in a bottom-up solvent evaporation approach and can have light weight, tunable optical properties and mild synthetic conditions (Fig. 1). A modular template approach is developed to mitigate the trade-off between porosity and glass forming ability, where the modulator would stabilize pore in the as-synthesized glasses even the linker is highly flexible. We demonstrate this chemistry using two linkers with different rigidity: the rigid linker methanetetrayltetrakis (benzene-4,1-diyl) tetramethanol (MTBT) gives Al-MTBT monoliths with high porosity but low glass forming ability; the flexible linker bis(2-hydroxyethyl) terephthalate (BHET) gives Al-BHET glasses with well-defined glass transition but low porosity. With a bulky monodentate alcohol 1,1,1-triphenyl-2,5,8,11-tetraoxatridecan-13-ol (TPTO) introduced as modulator template, the Al-BHET-TPTO shows high porosity while maintaining high glass forming ability, which shows Brunauer–Emmett–Teller (BET) surface area up to 500 m$^2$/g and well-defined glass transitions. The TPTO modulator template prevents pore collapse during the vitrification process and can be subsequently removed by solvent exchange to give gas accessible pores (Fig. 1e). The aluminum alkoxide glasses and monoliths are characterized by compositional, spectroscopic, calorimetric, mechanical, gas adsorption and X-ray scattering analysis, which reveals their structural features, porosity, optical properties and glass transitions. Comparison between the glass transition behavior, compositions and structure features of the as-synthesized and activated glasses shows that modulator molecules can have both pore-templating and plasticizing effect for the coordinatively linked network, which enable the combination of processability and porosity for the aluminum alkoxide glasses in a unique way: the as-synthesized glasses, with modulators in the pore,

show well-defined glass transitions and rheological behavior and can be processed like regular glasses; the activated glasses, with the modulator removed from the pore, show high gas uptake but absence of glass transition, and thus have fixed morphology and would not flow. We believe our findings provide a generalizable strategy to achieve a large library of highly porous glassy networks available for various applications.

## Results and Discussion

### Synthesis and characterization of Al-MTBT vitrified monolith

It is known that simple aluminum alkoxides can form AlOCs in solution, especially in the presence of carboxylic acid ligands[17–20]. In our synthesis, we aim to use multidentate alcohol linkers to link these AlOCs to form coordinative networks. The rigid MTBT linker is a promising candidate to produce porous networks in a vitrification process similar to that of titanium phenolate monoliths. For a typical synthesis, MTBT is reacted with aluminum sec-butoxide at 110 °C in a mixture of solvents consist of tetrahydrofuran (THF), ethylene glycol methyl ether and ethanol with the addition of glacial acetic acid to give a light-yellow transparent solution, which is then evaporated at 135 °C to give Al-MTBT as a vitrified monolith (Supplementary Figs. 1–13). In this process, the alcohol solvents have the same hydroxy group as the MTBT linker, which functions as a network modulator by coordinatively competing with MTBT and reducing the connectivity of the network. This process echoes the synthesis of titanium phenolate monoliths by evaporation of cresol solvent, where evaporation of coordinatively competitive solvent modulator would increase network connectivity and cause vitrification[15]. The as-synthesized Al-MTBT is a brittle solid that shows slight elasticity and its modulus can be measured by rotational rheometer in oscillation mode (Supplementary Fig. 9). The specific mixture of solvents is necessary to prevent precipitation of MTBT linker or amorphous coordination polymer. The synthesis of Al-MTBT also needs to be carried out with rapid evaporation; otherwise, precipitation of MTBT may occur. The Al-MTBT is mostly transparent

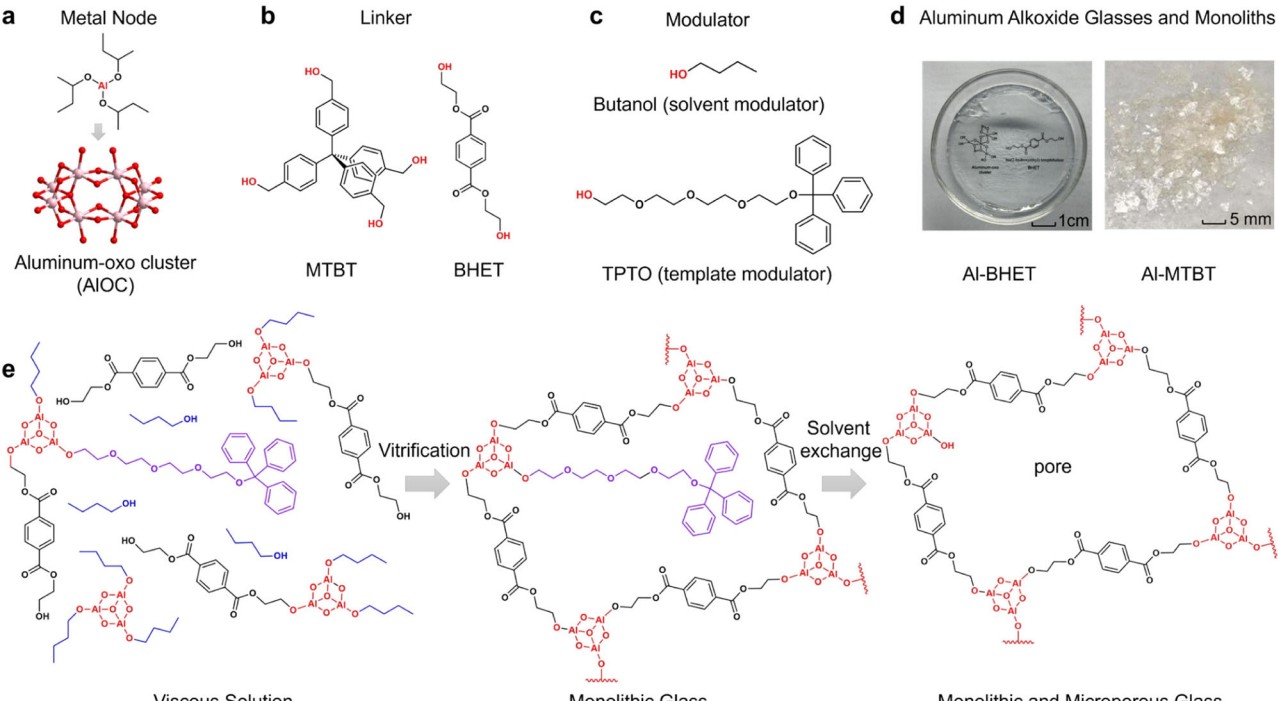

**Fig. 1 | The synthesis of aluminum alkoxide glasses and monoliths. a** In situ formed aluminum-oxo clusters (AlOCs) as the metal nodes. **b** Chemical structures of the two multidentate alcohols used as the organic linkers. **c** Chemical structures of the monodentate alcohols used as modulators. **d** Photographs of Al-BHET glasses and Al-MTBT monoliths made by linking the AlOC metal nodes with the corresponding linkers in the presence of butanol modulator. **e** The schematic showing pore expansion of Al-BHET glasses using TPTO as pore-templating modulator.

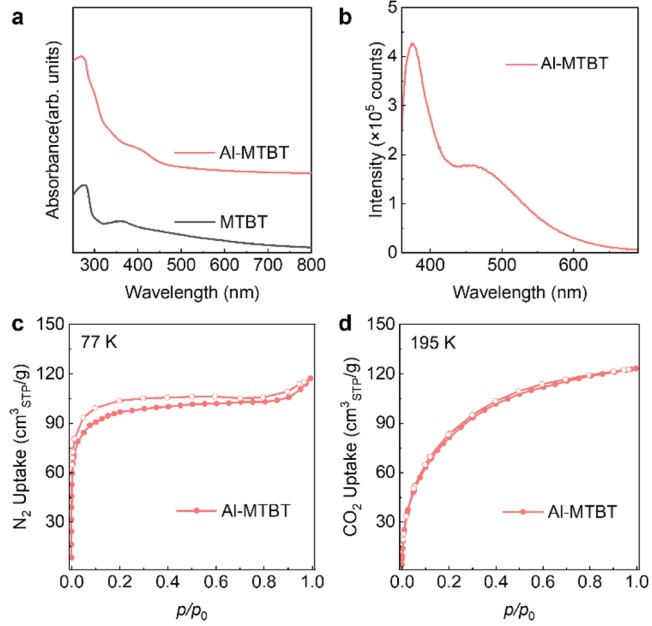

**Fig. 2 | Optical and porosity characterizations for Al-MTBT monolith.**
**a** Absorption spectra of Al-MTBT monolith and the MTBT linker. **b** Fluorescence spectrum of Al-MTBT when excited with 320 nm UV light. **c** Nitrogen sorption isotherm of Al-MTBT showing its gas accessible microporosity and surface area of 363 m²/g. **d** $CO_2$ uptake at 195 K for Al-MTBT. For **c** and **d**, the filled data points denote the adsorption curve and the hollow ones denote the desorption curve. Source data are provided as a Source Data file.

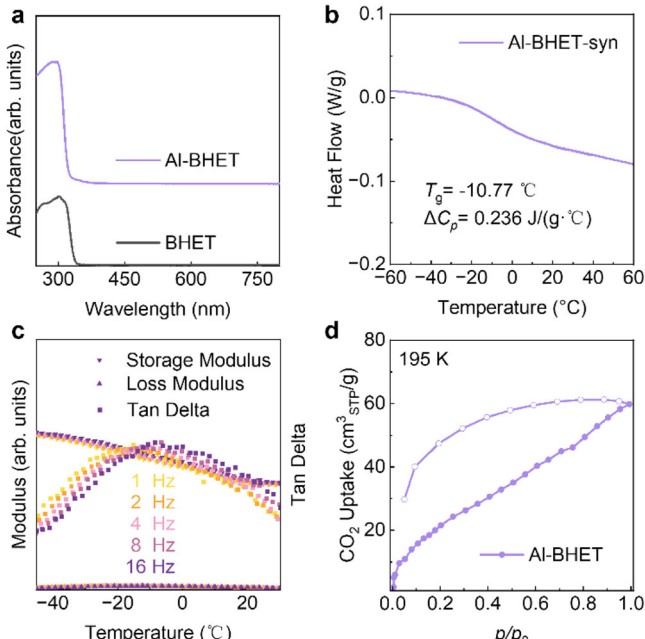

**Fig. 3 | Optical, glass transition and porosity characterizations for Al-BHET glasses.** **a** absorption spectra of Al-BHET and BHET linker showing their high transparency in the visible range. **b** Differential scanning calorimetry measurement of as-synthesized Al-BHET (denoted as Al-BHET-syn) under standard scan rate of 10 K/min showing its well-defined glass transition. **c** Dynamic mechanical analysis using a powder measurement kit showing the softening of as-synthesized Al-BHET glass near $T_g$, which indicates its configuration motion involves the overall framework. The measurement frequencies are varied from 1 Hz to 16 Hz, which are denoted by different colors from yellow to purple. The storage modulus, loss modulus and tangent delta are denoted by inverted triangle, triangle and square, respectively. **d** $CO_2$ uptake at 195 K of Al-BHET showing its gas accessible porosity with a pore volume of 0.11 cm³/g. The filled data points denote the adsorption curve and the hollow ones denote the desorption curve. Source data are provided as a Source Data file.

in visible range, with the MTBT show slight adsorption near 400 nm (Figs. 1d and 2a). Al-MTBT emits blue fluorescence under UV light, which originates from the linker (Fig. 2b).

Both optical images and scanning electron microscopy images of the Al-MTBT monolith show smooth surfaces and homogeneous features (Supplementary Fig. 2). The as-synthesized Al-MTBT is washed with THF and acetone and then activated with supercritical $CO_2$ drying and heating under vacuum. Al-MTBT monolith has high BET surface area of 363 m²/g measured by $N_2$ uptake at 77 K, a skeleton density of 1.33 g/cm³ and a pore fraction of 23% (Fig. 2c, d, Supplementary Fig. 3; Supplementary Tables 2–3). The porosity of Al-MTBT is also characterized by $CO_2$, and the presence of pores is supported by methanol and water uptake (Supplementary Figs. 4–6). The activated Al-MTBT also shows high thermal stability in thermalgravimetric analysis, and the absence of weight loss below 150 °C confirms that there are no guest molecules in the coordinative framework (Supplementary Fig. 7). The successful synthesis and high porosity of Al-MTBT monolith demonstrates the great potential of aluminum alkoxides in making porous monolithic networks.

Notably, in the initial report of monolithic titanium phenolate[15], their glassy nature was justified by increased glass transition temperature ($T_g$) with decreased modulator amount and making the analogy to highly cross-linked networks that do not show glass transition[15,21]. In this regard, the Al-MTBT is a glassy material similar to those titanium phenolates. However, as glass transition was not directly observed for Al-MTBT and their glassy nature is still under debate, we would denote them as vitrified monolithic networks for now. The lack of well-defined $T_g$ and sensitivity to specific synthetic conditions of Al-MTBT shows rigid linkers such as MTBT have fundamental limitations in producing coordinative networks with high glass-forming ability. Thus, it would be natural to pursue coordinative glasses that can combine high glass-forming ability and porosity with linkers that have high flexibility and solubility.

## Synthesis and characterization of Al-BHET glasses

BHET has higher flexibility and solubility compare to the MTBT linker, which is promising for designing aluminum alkoxide networks with higher glass-forming ability. The Al-BHET glasses can be made in a similar method as Al-MTBT monolith, where BHET, aluminum sec-butoxide and acetic acid are stirred in alcohol solvents at elevated temperature followed by evaporation induced vitrification (Fig. 1d, Supplementary Figs. 14–25). The Al-BHET is colorless and has no measurable absorption in the visible range of 400–750 nm (Fig. 3a). Due to the high flexibility of BHET linker, the as-synthesized Al-BHET shows well-defined $T_g$ around −10 °C, which is also supported by dynamic mechanical analysis (DMA) that shows sample softening (Fig. 3b, c). It is worth noting that the the absolute values of the modulus largely reflect the mechanical properties of the metal powder kit and could not be analyzed quantitatively. The Al-BHET at room temperature has the modulus of an elastic solid as evidenced by nano-indentation and rheology measurements (Supplementary Figs. 21, 22). The Al-BHET glasses can be activated in a similar manner as Al-MTBT, which involves washing with THF and acetone, super-critical $CO_2$ exchange and evacuation. Infrared spectroscopy reveals the absence of free BHET linker and indicates the deprotonation of the linker hydroxyl groups in the activated Al-BHET (Supplementary Fig. 23). The activated Al-BHET shows pore volume of 0.11 cm³/g determined by $CO_2$ uptake at 195 K, and its small pores give no nitrogen uptake at 77 K (Fig. 3d, Supplementary Fig. 14). The methanol and water uptake at room temperature also support the porosity of the material

(Supplementary Figs. 15, 16). The skeleton density of Al-BHET is determined by a He pycnometer to be 1.41 g/cm³, which is approximately half of the value of titanium phenolate glass (Supplementary Table 2)[15] From the pore volume measured by $CO_2$ uptake and skeleton density, the volumetric porosity of Al-BHET is determined to be 13%. The activated Al-BHET shows no weight loss below 150 °C, indicating the absence of unbounded guest molecules in the pore (Supplementary Fig. 17).

The structure of Al-BHET glass is analyzed by a combination of compositional analysis, spectroscopy, and total scattering techniques. In this process, we focus on describing short- and medium-range orders, which are essentially dictated by the structure of the AlOCs. As AlOCs are formed in situ, a variety of AlOCs are certainly present, and we aim to find a representative AlOC that embodies the most prominent features of these glasses. With the synthesis carried out in the presence of alcohol and carboxylic acid, the resulting AlOCs would have mixed acid and alcohol linkers. X-ray total scattering measurements produce the pair-distribution function of Al-BHET, where the first two peaks at 1.4 and 1.9 Å are attributed to C-C/O and Al-O bonds, respectively (Fig. 4a, b). The peak at 2.83 Å corresponds to the Al...Al pair, which clearly shows the presence of AlOCs. The absence of a peak at 3.3 Å excluded the presence of wheel-shaped AlOCs because their carboxylate groups with $\mu_2$-$\eta^1$:$\eta^1$ coordination mode would results in an Al-O pair at 3.3 Å[17–19]. In addition, these wheeled-shaped clusters typically have low solubility and can precipitate from the solution. On the other hand, AlOC-41 could be a more plausible candidate for representing the AlOCs in Al-BHET[20], which incorporates only a small number of bridging carboxylic acids. The AlOC-41 cluster also has a higher solubility and can be more readily formed under the synthesis conditions used to prepare Al-BHET. The pair-distribution function simulated using the core of AlOC-41 attached to the BHET linker show good consistency with measured PDF data. Notably, most of the peaks in the pair-distribution function are associated with various atom pairs, and the allocation in Fig. 4a does not suggest that the corresponding peaks originated exclusively from these pairs. In addition, the 2.36 Å peak does not heavily involve Al and thus would primarily originate from the linker; this peak is not shown in Fig. 4a for simplicity. Considering the relatively high acid-to-alcohol ratio of 2.25 found via magnetic resonance spectroscopy (NMR) for the digested sample (Supplementary Fig. 24), part of the carboxylic acid would be a terminal ligand that replaces BHET. This provides the empirical formula of $Al_8O_2(OH)_3(BHET)_{4.2}(CH_3COO)_{9.6} \cdot 2H_2O$, with a calculated composition of C 43.0% H 4.4% O 41.5% Al 11.1%, which is consistent with the measured elemental analysis result of C 43.1% H 4.9% O 42.8% Al 10.3%. X-ray absorption spectroscopy (XAS) also provides corroborating evidence for the presence of AlOCs. As AlOC-41 is unstable at ambient moisture and can degrade during sample installation at the synchrotron beamline, another stable and synthetically viable AlOC, AlOC-12, was used as the reference sample in XAS. The AlOC-12 and AlOC-41 have similar structures in terms of aluminum coordination environment (octahedral), types of ligands (mixed carboxylates, alkoxides and bridging hydroxides) and Al-O bond lengths. The near edge features of the Al-BHET were similar to a prototype AlOC-12 and distinctly different from alumina (Fig. 4c)[17,22,23]. Qualitatively, the R space plot for Al-BHET and AlOC-12 shows that the Al-O bonds in these two materials have similar length, which is consistent with presence of AlOCs incorporating alcohol and carboxylic acid ligand in Al-BHET (Supplementary Fig. 25). Notably, as the Al-BHET is highly insulating, the signal-to-noise ratio in the extended X-ray adsorption fine structure regime is limited, which prevents reliable analysis beyond first shell or quantitative fitting for the aluminum coordination environment. Solid-state ²⁷Al cross-polarization magnetic angle spinning (CP-MAS) NMR with shows the aluminum in Al-BHET is in an octahedral coordination environment, consistent with the structural model of AlOC-41 (Fig. 4d)[24]. The consistency of the above analysis shows that it is reasonable to speculate that the aluminum alkoxide glasses are composed of AlOCs linked by alcohol linkers and reinforced by carboxylic acid ligands, where the AlOCs would have mixed ligands and similar types and sizes of AlOC-41.

Synthesizing composite materials with crystalline and glassy network can be a viable way to achieve high surface area monolithic material[25,26]. As a proof-of-concept, UiO-66 nanocrystals were added to the solution of Al-BHET synthesis to give UiO-66@Al-BHET composite that shows high surface area of 680 m²/g (Supplementary Figs. 26–28).

Comparing Al-BHET and Al-MTBT, it is clear that there is a trade-off for glass forming ability and porosity, where more flexible linkers such as BHET tend to give higher glass forming ability but lower porosity. Comparing the as-synthesized and activated Al-BHET has provides hint to mitigating such trade-off. In the activation process, the residue solvent-modulator and unbounded BHET linker are removed from the pore, which makes the pore accessible to $CO_2$ gases and result in disappearance of $T_g$ in the differential scanning calorimetry (DSC) curve (Supplementary Fig. 20). Such observation indicates that modulator molecules in the pore could both stabilize the pore and facilitate configurational motion of the framework. It is thus natural to speculate that modulators bearing the same hydroxyl group as butanol and BHET but have larger sizes can give high porosity while preserving the glass forming ability of Al-BHET.

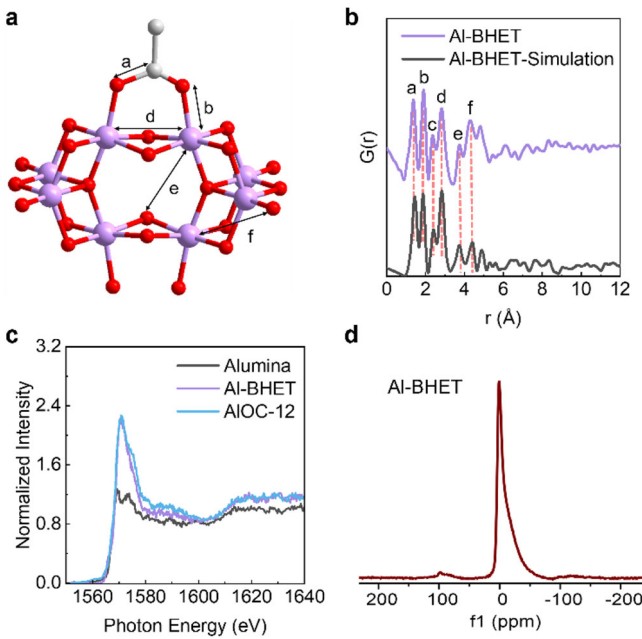

**Fig. 4 | Structure elucidation of Al-BHET glasses. a** The inorganic core of AlOC-41 which is used as the AlOC to model the short- and medium range structure of Al-BHET. The prominent atom pairs of this cluster is labeled for comparison with the pair-distribution function. Al, purple spheres; C, gray spheres; O, red spheres. **b** experimental pair-distribution function and the pattern simulated using the AlOC-41 as the metal node showing good agreement. The prominent features in the pattern are labeled in correspondence to the atom pairs in **a. c** X-ray absorption spectrum of Al-BHET supporting the presence of AlOCs, which shows similar features as the reference AlOC-12 and are distinctively different from alumina. **d** The solid state ²⁷Al CP-MAS NMR showing the Al atoms in Al-BHET are in octahedral geometry, which is consistent with the AlOC structure. Source data are provided as a Source Data file.

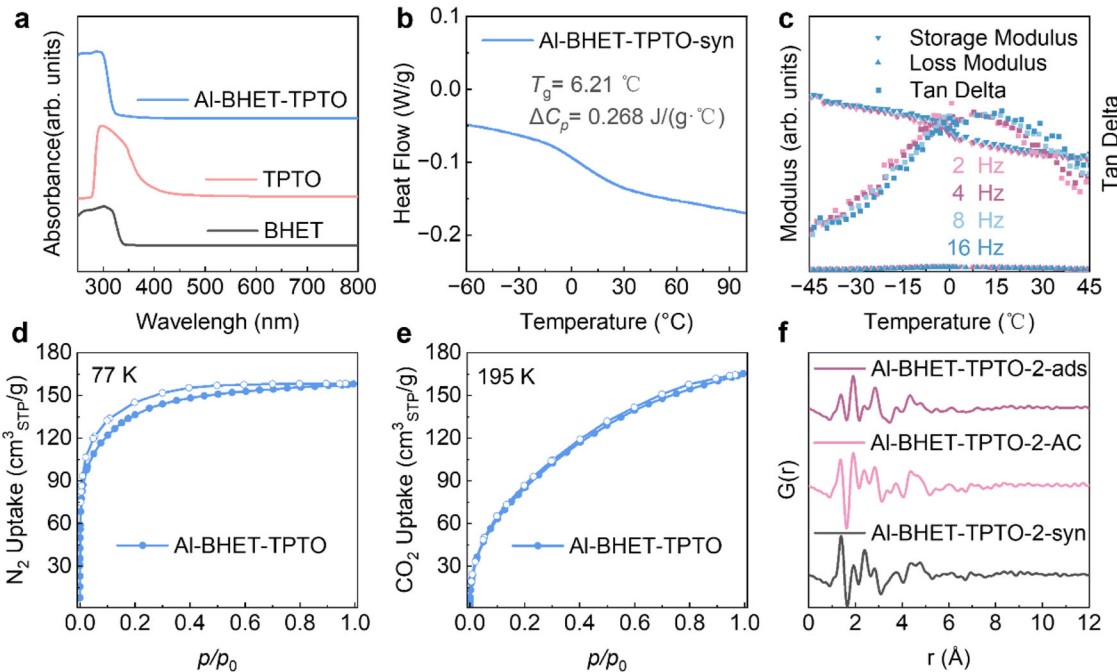

**Fig. 5 | Optical, glass transition, porosity and structural characterizations for Al-BHET-TPTO glasses. a** Absorption spectra of Al-BHET-TPTO glasses, BHET linker and TPTO modulator showing their high transparency in the visible range. **b** Differential scanning calorimetry measurement of as synthesized Al-BHET-TPTO (denoted as Al-BHET-TPTO-syn) under standard scan rate of 10 K/min showing its well-defined glass transition. **c** Dynamic mechanical analysis using a powder measurement kit showing the softening of as-synthesized Al-BHET-TPTO glass near $T_g$, which indicates its configuration motion involves the overall framework. The measurement frequencies are varied from 2 to 16 Hz, which are denoted by different colors from pink to blue. The storage modulus, loss modulus and tangent delta are denoted by inverted triangle, triangle and square, respectively. **d** Nitrogen adsorption isotherm of the activated Al-BHET-TPTO glass showing a surface area of 500 m²/g. **e** $CO_2$ uptake of the activated Al-BHET-TPTO glass. For **d** and **e**, the filled data points denote the adsorption curve and the hollow ones denote the desorption curve. **f** Pair-distribution functions of as-synthesized, acetone exchanged, and activated Al-BHET-TPTO-2, (denoted with suffix of -syn, -AC and -ads, respectively). The Al-BHET-TPTO-2 has higher TPTO content in the as-synthesized form, which can more clearly show the the mild activation process largely preserve the local structure of the coordinative backbone. Source data are provided as a Source Data file.

## Synthesis and characterization of Al-BHET-TPTO glasses with modulator template

Compared to solvent modulators such as butanol, nonvolatile bulky monodentate alcohols are more suitable for creating high porosity in aluminum alkoxide glasses, as they have larger sizes and can be precisely quantified in the synthetic mixture. During solvent evaporation induced glass synthesis, these modulators can coordinate to AlOCs and prevent pore collapse. After vitrification, the modulators can be removed by solvent exchange with short alcohols to provide micropores. These modulators, bearing the same hydroxy functional groups as the linker, could also undergo coordination exchange with the linker and facilitate configurational motion of the framework, thus act as plasticizer and improve glass forming ability.

We demonstrate such an approach by using TPTO as a modulator (Supplementary Figs. 29–42). The Al-BHET-TPTO glass is synthesized with 23% TPTO (molar ratio compared to BHET) initially added to the solution, which is also optically transparent glassy and shows well-defined glass transition near 6 °C in its as-synthesized form (Fig. 5). After solvent exchange with THF/ethanol mixture and acetone, digested NMR and infrared spectroscopy show that only a small amount of TPTO is left in the activated glass (BHET:TPTO = 11.7) compared to the amount initially added (Supplementary Figs. 38, 41). The activated Al-BHET-TPTO glass show higher porosity compared the Al-BHET as measured by nitrogen uptake at 77 K and $CO_2$ uptake at 195 K (Fig. 5d, e), which show a surface area of 500 m²/g with a void fraction of 30%. With most TPTO molecules in the pore removed, the activated glass no longer show $T_g$ in the DSC measurement, indicating the TPTO also act as network plasticizer similar to the alcohol solvents.

The DMA measurement shows the softening of Al-BHET-TPTO around $T_g$, which indicates the configurational motion associated with glass transition involves the backbone dynamics of the overall framework and not solely due to the motion of guest molecules in the pore (Fig. 5c). It is worth noting that the surface area and glass transition temperature of the Al-BHET-TPTO glass does not show monotonic dependence on the BHET:TPTO ratio, as the modulator affects the network structure and dynamics in a rather complex manner (Supplementary Figs. 43–56).

Notably, the mild activation process is entirely carried out at room temperature and only involves solvent exchange, super critical $CO_2$ drying, and evacuation, which is adopted from the activation of delicate MOFs and would only remove guest molecules in the pore without affecting the coordinatively linked backbone[27]. The preservation of the coordinatively linked aluminum alkoxide backbone during activation is confirmed by the X-ray total scattering and solid-state NMR, where the prominent features of the pair-distribution functions remain largely unchanged for the as-synthesized, solvent-exchanged, and activated Al-BHET-TPTO glasses even the TPTO:BHET ratio is increased to 1:1 in the initial synthesis (Fig. 5f). The $^{27}$Al CP-MAS NMR also shows the octahedral coordination environment is preserved for the activated glasses. Thus, the activated glasses, although without well-defined glass transition, are structurally identical to the as-synthesized glasses in terms of the coordinatively linked backbone, and the only difference between them is whether the pore is filled with modulator molecules. Consequently, the activated glasses also inherit the transparency and homogeneity of the as-synthesized glasses. From the perspective of the as-synthesized

glasses, the activation and gas adsorption process merely exchange the modulator in the pore with gases. As the term "pore" means the voids of a robust framework where guest molecules can move in and out, the as-synthesized glass should also be considered as porous. However, the porosity of the aluminum alkoxide glasses is fundamentally different from the porosity of zeolitic imidazolate framework glasses, which is open to guest molecules immediately after quenching from the super-cooled liquid without the need for any additional activation process. From a practical perspective, the aluminum alkoxide glasses can be first shaped in the as-synthesized form and then activated, thus combining the processability of glasses, porosity of crystalline frameworks and the modular designability of reticular chemistry.

In summary, we developed a series of metal-organic network-forming glasses and monoliths made from AlOCs and alcohol linkers. Its synthesis can be carried out via a bottom-up approach, where evaporation of coordinatively competitive alcohol solvents leads to increased network connectivity and subsequent vitrification. The introduction of a bulky network modulator TPTO produces a porous glassy coordinative network Al-BHET-TPTO that shows well-defined glass transition, and its configurational motion at $T_g$ corresponds to overall framework motion facilitated by TPTO. The plasticizing effect of TPTO is associated with its hydroxy functional group that coordinatively compete with the BHET linker. The TPTO in the pore can be removed in a mild activation process that preserves the porous backbone of the as-synthesized glasses, which show high porosity as determined by gas adsorption isotherm. Such modulator template approach enables the modular synthesis of highly porous coordinative network glasses with flexible linker, which can have high glass forming ability, optical transparency and mild synthetic conditions. The development of aluminum alkoxide glasses and modulator templating strategies represents an important advance in the field of metal-organic network-forming glasses, which have expanded the structural diversity of these materials and provided viable solutions for achieving high porosity.

## Methods

### Instruments and methods
Differential scanning calorimetry is measured with TA DSC250 instrument under nitrogen atmosphere. PerkinElmer Dynamic Mechanical Analyzer DMA 8000 was used for DMA, using single cantilever-rectangle measuring system under a nitrogen atmosphere of 20.0 mL/min, and heating rate of 2.00 °C/min. The static force is 2.00 N, the force multiplier is 1.1, and the strain is 0.005 mm. NMR is measured by a Bruker AVANCE III HD500 spectrometer. A solvent mixture of DMSO-$d_6$ and $D_2O$ with the addition of KOH was used to digest the aluminum alkoxide glasses. The ADVANCE III HD 400 MHz solid-state NMR spectrometer from Switzerland Bruker BioSpin AG and 3.2 mm CP-MAS room temperature probes were used for $^{27}$Al at a speed of 10 kHz. Pair-distribution function is measured with a Rigaku Smartlab Studio II instrument with silver targets, where the scanning range is 3-157° and the scanning speed is 0.1 °/min. XAS was measured at BL02B of Shanghai Light Source with alumina as a calibration standard. The sample activation was carried out with Tousimis Samdri-795 critical point dryer and Quantachrome Instruments' Anton Paar brand FloVac degassing station. A MicrotracBEL Belsorp-MAXII high-throughput specific surface area analyzer was used to perform gas and vapor adsorption measurement.

### Synthesis of Al-BHET
310 mg of BHET (1.22 mmol) is dissolved in 2 mL 1-butanol, 500 μL glacial acetic acid, and 4 mL ethanol in a glass vial placed on a hot plate heated to 120 °C. Then 300 mg aluminum sec-butoxide (Al-(O*s*Bu)₃, 1.22 mmol) is added to this solution, which is stirred overnight on the 120 °C hotplate to get a clear, colorless solution before evaporation in a petri dish pre-heated on a 125 °C hotplate to give the Al-BHET glass.

### Synthesis of Al-BHET-TPTO
To 495 mg (1.95 mmol) of BHET and 195 mg (0.45 mmol) of TPTO, 8 mL ethanol, 4 mL 1-butanol, and 1 mL glacial acetic acid are added to give a clear and colorless solution, which is then added to 495 mg (2.01 mmol) Al-(O*s*Bu)₃ and stir overnight at 120 °C hotplate. The Al-BHET-TPTO glass is obtained by evaporating the solution in a petri dish on a hotplate set at 128 °C.

### Synthesis of Al-BHET-TPTO-2
To 207.2 mg (0.815 mmol) of BHET and 355.8 mg (0.815 mmol) of TPTO, 4 mL ethanol, 2 mL 1-butanol, and 500 μL glacial acetic acid are added to give a clear and colorless solution, which is then added to 300 mg (1.22 mmol) Al-(O*s*Bu)₃ and stir overnight at 120 °C hotplate. The Al-BHET-TPTO glass is obtained by evaporating the solution in a petri dish on a hotplate set at 128 °C.

### Synthesis of Al-MTBT
4 mL tetrahydrofuran, 4 mL ethylene glycol methyl ether, 2 mL ethanol and 100 μL glacial acetic acid is added to 120 mg (0.27 mmol) of MTBT to give a clear solution. The solution was then added to 276 mg (1.12 mmol) Al-(O*s*Bu)₃ and stirred overnight at 110 °C to give a light-yellow transparent solution. The solution is then evaporated in a petri dish on a hotplate at 135 °C.

### Activation
All aluminum alkoxide glass samples were activated by solvent exchange followed by super-critical CO2 drying. Al-BHET and Al-MTBT were exchanged three times with tetrahydrofuran, followed by three times with acetone. The Al-BHET-TPTO samples were exchanged with a 1:1 mixture of tetrahydrofuran and ethanol for 4 days, followed by 3 exchanges with acetone. The composite material is directly vacuum-dried after three exchanges of tetrahydrofuran. All samples were degassed for 2 hours using a degassing station before gas adsorption testing.

## Data availability
The data that support the findings of this study are available from the corresponding authors upon request. All data are available in the main text or the supplementary materials. Source data are provided in this paper.

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

## Acknowledgements

The authors thank Ms. L. Long at ShanghaiTech University for assistance in gas adsorption measurements, Dr. R. Gao and Mr. C. Qi for assistance in fluorescence measurements, Ms. R.-F. Ma and Prof. M.-H. Zeng at Guangxi Normal University for X-ray total scattering measurements, Prof. W. Fang for providing AlOC-12 as reference for XAS, Prof. T.-C. Weng and Dr. H.-B. Dou for discussion on the XAS data. X-ray absorption spectroscopy at the Al K-edge was carried out at BL02B02 of the Shanghai Synchrotron Radiation Facility, which was supported by the ME2 project under contract from the National Natural Science Foundation of China (11227902). Y.Z. acknowledges support from the Science and Technology Commission of Shanghai Municipality (22QC1401500) and start-up funding from ShanghaiTech University.

## Author contributions

Y.Z. and Z.Z. initiated the research project, conducted the experiments, organized the result, and wrote the manuscript.

## Competing interests

The authors declare no competing interests.
