## [Peer Review File · Nature Communications]

Transparent and High-porosity Aluminum Alkoxide Network-forming GlassesREVIEWER COMMENTS

Reviewer #1 (Remarks to the Author):

This study presents the direct synthesis of an amorphous metal-organic framework with glass transitions from an aluminium cluster with alcohol ligands, following the methodology outlined in a prior publication by one of the authors (J. Am. Chem. Soc. 2016, 138, 34, 10818–10821). One of the interesting aspects of this work is the approach to maintaining permanent porosity using modulator templates, which have not yet been presented in MOF glass design. However, the current manuscript raises several concerns, and a major revision is necessary for further consideration. I have listed my concerns and suggestions below:

1. One major concern in the manuscript relates to the terminology used to describe the materials. The authors refer to their amorphous samples as "glass" throughout the manuscript. However, direct evidence of having an actual glass transition under the DSC of the amorphous MOF itself remains unclear. Overall, only samples with the TPTO modulator show glass transition temperatures. While the authors provide justification in the "The chemical space of Al/BHET/TPTO and the glass transition of aluminium alkoxide glasses" section (lines 150–184), it remains uncertain whether the observed heat flow in DSC measurements is attributed to the relaxation of the $n-(O-CH_2-CH_2)_n-$ of TPTO molecules inside the pore rather than that of the entire structure. Similar behaviour has been observed in other framework materials, such as when glassy poly(ethylene oxide) is anchored within a covalent organic framework (J. Am. Chem. Soc. 2019, 141, 3, 1227–1234). This analogous relaxation phenomenon in DSC measurements, however, does not categorise the crystalline COF as glass.

Further characterizations, including DSC and dynamic mechanical analysis (DMA) or thermal mechanical analysis (TMA), of samples without TPTO (Al-BHET, Al-MTBT) and samples with TPTO after activation (similar to the condition for gas adsorption measurement) to clarify the glass transition behavior are essential.

2. In the "High-porosity MOF@glass composites" section (lines 222–229), the term "glass ceramics" is inaccurately used. The term "composites" seems to be more suitable in this case.

3. EXAFS and XAS discussions (lines 214–215) are unclear. How can the presence of AlOC be confirmed by comparing EXAFS and XAS with alumina? In addition, I recommend adding actual EXAFS and XAS data for alumina to Figures 5C, S6, and S7. Is Figure 5C identical to Figure S6?

4. The thermal stability of all samples should be discussed.

5. Figure 6b requires the isotherm of pristine UiO-66, and the mass loading of UiO-66 should be described in the manuscript.

6. I suggest using a linear scale for the x axis in all pair-distribution function data.

Overall, the quality of the figures, terminology, discussions, figure captions, and data interpretations are unclear. Throughout the manuscript, several explanations and discussions are mentioned without appropriate citation or direct evidence. A careful revision and additional experiments are necessary.

Reviewer #2 (Remarks to the Author):

In this manuscript, Prof. Zhao and his colleague reported a type of new glass forming metal organic glass materials, fabricated through templated removal of the organic solvent. The method is versatile and the resultant materials can preserve a large amount of porosity. This materials is one of the more porous glass reported in the literature and it would find good use in many applications. The materials characterisation is clearly in depth - with advanced characterisation techniques including x-ray absorption spectroscopy and total scattering, the atomic pair distribution and fine structure can be retrieved, which further shed lights onto the arrangement of the metal-organic hybrid materials. The authors have also demonstrated the capability of compositing with other functional materials like UiO66. Overall, I think this paper is of high quality and can be published in nature communication after some minor revision.

1. The author states the materials features good glass forming capability - would heating the liquid MOF to a higher temperature and then quenching back to RT lead to any re-crystallisation?
2. Is it possible to provide some data on the mechanical properties of these materials? Given the T_g is around RT range, would that means the materials behave more like a gel under ambient?
3. The large hysteresis effect as evidenced by the gas isothermo need further discussion and clarification, as this may affect the practical use of the highly porous materials.

Reviewer #3 (Remarks to the Author):

This manuscript proposes a strategy for synthesizing coordination network glasses with high internal surface. The authors have synthesized two kinds of metal-organic glasses using a solvent evaporation approach. To synthesize Al-BHET and Al-MTBT glasses, they dissolved aluminum sec-butoxide together with BHET or MTBT into glacial acetic acid/butanol/ethanol or ethylene glycol methyl ether/THF/ethanol solutions, respectively. The resultant solutions were heated at 125 °C to evaporate the solvent solutions and obtain glassy samples which demonstrate gas adsorption capability and low glass transition temperatures (below 0°C). They also incorporated TPTO into the Al-BHET and Al-MTBT glass networks as a modulator which can be removed by alcohol exchange and finally increase the porosity of glass. The prepared glasses possess high porosity. It is an interesting work for the structure control and porosity design of coordination glasses. Yet, the local structure and chemical compositions of the prepared glasses is not clear enough. Numerous issues need to be addressed. Due to the low T_g values (below 0 degree), the application of these glasses at ambient conditions will be limited. This manuscript requires major revisions before potentially being published

in Nature Communications. My specific comments and suggestions for improving the manuscript are as follows:

1. The authors used metal:organic linker molar ratios of 1:1, 1:1.2, and 4.15:1 for the preparation of the Al-BHET, Al-BHET-TPTO, and Al-MTBT glasses, while there are no experimental procedures for the preparation of Al-MTBT-TPTO glass. Why did the authors use these molar ratios? What is the chemical composition of each glass? How does this metal:organic molar ratio influence the glass properties in terms of porosity and glass transition?
2. The authors used solvent exchange method to remove TPTO by THF and acetone. Based on NMR and FTIR data, they claimed that only a very small amount of TPTO is left in the activated glass compared to the amount initially added. However, the FTIR data confirm that a large amount of TPTO is left in the glass and BHET which dissolved during solvent exchange. The intensity of the resonance peak at 1100 cm⁻¹ (TPTO) increases, while the intensity of the resonance peaks of BHET decreases after solvent exchange as shown in Figure S19.
3. The authors mentioned that their glasses are the most porous glassy materials with high surface areas up to 500 m²/g. This is incorrect, there is another metal-organic glass based on titanium clusters which has been reported with surface areas of 923 m²/g which is much higher than that reported in this manuscript (Xu, W. et al. High-Porosity Metal-Organic Framework Glasses. *Angew. Chem. Int. Ed.* 62, 278 e202300003, 2023).
4. As shown in Figure 1, the authors mentioned that aluminum sec-butoxide reacts with BHET/TPTO or MTBT/TPTO during solvent evaporation at 125 °C, forming coordination bonds such that aluminum cluster bridges two BHET or MTBT. Could you provide evidence for this assumption?
5. The simulated and experimental PDF data show the short-range atom-atom correlations in the aluminum clusters. As is known, ZIF-glasses show a high degree of short-range disordered structure due to the weak coordination bonds and large ligands (see Madsen et al, *Science* 367 (2020) 1473-1476). The authors should describe the short-range structure in their glasses in terms of disorder or order. From Fig. 1, it seems that the medium range structure of your glasses is of high degree of order. The authors should comment on this.
6. Furthermore, the FTIR data strongly suggest that the coordination bonds are not formed in the prepared glasses. The resonance peak of OH group in the BHET/MTPT linkers at 3500 cm⁻¹ is clearly observed in all the spectra of the prepared glasses as shown in Figure S4,S13,S19,S25. This means that the BHET/MTPT linkers have not deprotonated during solvent evaporation. The authors also should provide the FTIR spectrum of the aluminum sec-butoxide for comparison to clearly see what happened between aluminum clusters and BHET/MTPT linkers during solvent evaporation.
7. In my opinion and based on the performed structural characterizations for the prepared glasses, the dissolution of aluminum alkoxide and BHET/MTPT into alcohols (e.g., THF, ethanol, butanol) followed by thermal treatment at 125 °C, results in the solvent evaporation and the formation of 3D hydrogen bonded molecular glass network. To some extent, this glass has some similar features to the previously reported metal inorganic-organic hybrid glass with supramolecular network (Ali, et al, *Angew. Chem. Int. Ed.* 2023, 62, e202218094). In this glass, aluminum clusters, solvent, and BHET/MTPT/TPTO are connected with each other through hydrogen bonds. The organic linkers may coordinate with aluminum clusters forming large molecules but the coordination bonds do not form in this glass as they mentioned in their discussions. The coordination of organic linkers with

aluminum clusters should be examined carefully. Therefore, the structural related discussions should be revised. The origin of the porosity in the aluminium alkoxide glass is not clear. Is it due to the structural property or is it induced during solvent exchange process which results in the dissolution of glass constituents and form pores? The authors should examine the porosity and the composition of the glass samples especially before and after solvent exchange and activation process. This could help them to understand the origin of the porosity for their glasses.

8. In lines 57, 80 and 88, the authors claim that the viscosity of clear and colorless solution increase and then the vitrification occurs. However, the clarity of this process remains obscured solely by the authors' statements. Before activation, the product likely encompasses various components such as solvent molecules, by-products, and unreacted materials. How to prove the sample turns to glass rather than the very sticky solution? What is the product looks like? Is it hard or soft? Given the imperceptible glass transition of Al-BHET and Al-MTBT glass, it prompts consideration of whether the products synthesized are well-defined glass or only amorphous materials.

9. In lines 96 and 98, as well as in Figure S2, the authors mention the possibility of crystallization occurring during the preparation process. The SEM images with TPTO modulator also reveals heterogeneous shapes. Are these morphologies indicative of crystallization, or do they signify the presence of impurities? Further clarification is needed for the crystallization that produced during the preparation process.

10. In line 109 and Figure S3, the author attempted to demonstrate the porosity of the product through a methanol adsorption test. However, it is worth considering that the glass may harbor numerous reaction points with methanol, leading to the connection of gas molecular and glass structure rather than being absorbed within the pore of glass structure.

11. In the section discussing "chemical space and glass transition," the author used DSC to analysis variation in glass transition with different TPTO ratios. However, according to the supporting information, it remains unclear whether the test samples underwent an activation process. If the response is negative, the presence of residual TPTO could potentially impact the results, leading to a gradual attenuation of the glass transition signal. This scenario poses a challenge in conclusively establishing the glass transition nature of the samples.

12. In the left panel of Figure 4, please indicate where T_g is and how to determine the T_g values. Please describe the conditions for the DSC measurement, e.g., upscan and downscan rates, atmosphere, pressure. The standard T_g values should be measured under the standard conditions (Zheng et al. Chem. Rev. 119 (2019) 7848-7939). Fig. 4: Please label a) and b) inside the figure.

13. Since the T_g values are below 0 degree, what kind of applications do you expect?

14. Figures S4, S19 and S25 show the FTIR spectra of Al-BHET, Al-BHET-TPTO and Al-BHET-TPTO-2. The syn samples exhibit a good alignment, whereas the ads samples do not match as closely with each other. What makes the difference? Is it possible that the solvent exchange process damages the sample's structure? Alternatively, could varying TPTO ratios lead to structural changes in the glass after activation?

15. In Figure S24, there is a noticeable alteration in the PDF diagram following various solvent exchange processes. The question arises: is this change attributable to an influence of different solvent on the glass structure, or does it suggest the penetration of solvent into the glass structure?

16. The section on "High-porosity MOF@glass composites" appears brief and simplistically

described. SAED and EDS analyses could provide a more comprehensive understanding of the composite structure. Elaboration on how these components interconnect and an exploration of the impact of the glass's chemical and physical properties on UiO-66 would contribute to future's work.

17. There are some grammars and writing problems (for instance, in line 92, "solvent mixture" should be amended to "mixture for solvent."). Please carefully review and polish the entire text.

18. The figures and their respective captions need to be improved to facilitate understanding for the authors.

Reviewer #4 (Remarks to the Author):

In this work, Zhang et al. developed a new series of aluminum alkoxide glasses through coordinatively competitive solvents evaporation. Meanwhile, they also provide a generalizable method to balance the porosity and glass forming ability of MOF glasses by using modulator templating approach. This strategy sheds the light for forming some porous MOF glasses with flexible ligands. But it's important to point out that this strategy needs more validation. For example, during the process of washing the sample to remove template molecules, will the ligand BHET be also washed away? How much TPTO remains in the final porous sample? This requires NMR and GC characterization of the sample after acid hydrolysis. DSC results in Fig4 imply that the TPTO content in the sample may be large. In other words, the porous skeleton in the Al-BHET-TPTO sample is actually constructed from BHET and TPTO. This is actually very contradictory to the porosity formation mechanism proposed by the author (Fig1e). By the way, some similar synthesis strategies have been already reported (Ref 15, 16), which reduces the novelty and importance of this work to some extent.

1. In Figure 1d, Al-BHET glass seems very optically transparent, but how about the transparency of Al-BHET-TPTO glass?
2. One of the indicators to determine the glass formation is whether the material is stable within the glass transition temperature (T_g) range. Could you provide more detailed thermal stability information of your glasses?
3. Structural analysis is very important in this work. I think the data quality of the XAFS results is very poor (Figure 5c, Figure S6 and Figure S8), which makes the current analysis results untrustworthy. If data quality can be improved, authors need to provide necessary data fitting.
4. For better comparison, it is better to provide the results of AIOC in both Figure 5b and Figure 5c.
5. Considering the complexity, is it possible for the author to detailly explain the process of obtaining simulation results of pair-distribution function again?
6. The peak intensity in Figure 6a is too low, so it is hard to check the purity and crystallinity of UiO-66@Al-BHET glass ceramic.
7. From the SEM image in Figure 6c, we can see the UiO-66 nanocrystals on the surface of Al-BHET glass. And I am curious about the cross-section image of this glass ceramic.
8. The work about preparing "high-porosity MOF@glass composites" broadens the potential

application field of Al-BHET glass, however, this part of content has little to do with the main content of the article. It is better for the author to move this part into the supporting information.

Reviewer #5 (Remarks to the Author):

The article by Z. Zhang and Y. Zhao offers an interesting new approach to metal–organic network-forming glasses with good porosity, via a bottom-up synthesis. The article is well written, and I believe that it demonstrates the novelty required to merit publication in Nature Communications, with minor revisions/clarifications.

Why does Al-MTBT display a moderate N₂ SA of 362.544 m²/g, yet a lower CO₂ pore volume than Al-BHET-TPTO-2, which displays the lowest N₂ SA?

The authors state that “The methanol uptake at room temperature also confirmed the porosity of the material”, though this statement may require a little more explanation. Are these materials stable to water vapour? I wonder if the authors have looked at the water uptake of the materials in a similar way to how methanol uptake was determined?

The authors note the formation of crystal glass composite materials using the highly porous UiO-66 as the crystalline phase. This approach is similar to several other studies, including Nat. Commun., 2019, 10, 2580 and Chem. Sci., 2020, 11, 9910-9918. I am not sure that this adds much to the study, and perhaps takes away from the interesting properties of the original glasses. Furthermore, the authors use the term “glass ceramics” – terminology with which I disagree. Glass ceramics are produced by selective crystallisation from a base glass, so these composites should be referred to as MOF-CGCs, not glass ceramics. The materials also should not be referred to as hybrid analogues of glass ceramics (Chem. Eur. J., 2022, 28, e202104026), as the crystalline and glass phases are from different materials.

REVIEWER COMMENTS

Reviewer #1 (Remarks to the Author):

This study presents the direct synthesis of an amorphous metal-organic framework with glass transitions from an aluminium cluster with alcohol ligands, following the methodology outlined in a prior publication by one of the authors (*J. Am. Chem. Soc.* 2016, 138, 34, 10818–10821). One of the interesting aspects of this work is the approach to maintaining permanent porosity using modulator templates, which have not yet been presented in MOF glass design. However, the current manuscript raises several concerns, and a major revision is necessary for further consideration. I have listed my concerns and suggestions below:

1. One major concern in the manuscript relates to the terminology used to describe the materials. The authors refer to their amorphous samples as "glass" throughout the manuscript. However, direct evidence of having an actual glass transition under the DSC of the amorphous MOF itself remains unclear. Overall, only samples with the TPTO modulator show glass transition temperatures. While the authors provide justification in the "The chemical space of Al/BHET/TPTO and the glass transition of aluminium alkoxide glasses" section (lines 150–184), it remains uncertain whether the observed heat flow in DSC measurements is attributed to the relaxation of the a $-(O-CH_2-CH_2)_n-$ of TPTO molecules inside the pore rather than that of the entire structure. Similar behaviour has been observed in other framework materials, such as when glassy poly(ethylene oxide) is anchored within a covalent organic framework (*J. Am. Chem. Soc.* 2019, 141, 3, 1227–1234). This analogous relaxation phenomenon in DSC measurements, however, does not categorise the crystalline COF as glass.

Further characterizations, including DSC and dynamic mechanical analysis (DMA) or thermal mechanical analysis (TMA), of samples without TPTO (Al-BHET, Al-MTBT) and samples with TPTO after activation (similar to the condition for gas adsorption measurement) to clarify the glass transition behavior are essential.

We thank the reviewer for raising this important concern. In fact, the glassy nature of the monolithic coordinative networks made in this bottom-up solvent evaporation approach have been subjected to debate. In the initial paper (*J. Am. Chem. Soc.* 2016, 138, 34, 10818), Austen Angell justify the glassy nature of the material by showing its increased T_g with decreased amount of solvent-modulator and making the analogy to highly cross-linked polymer:

Fig. R1. (a) the figure 4a in *J. Am. Chem. Soc.* 2016, 138, 34, 10818; (b) fig. 4. in the original manuscript (now removed in the revised manuscript)

“The Ti-BPA and Ti-BPP glasses, with 3D network structure, prove to imitate very dry silica, vitreous water in its low-density polyamorphic (LDA) form, and also most of the common ambers (highly cross-linked organic glasses from geologically distant times). These are all problematic because their T_g 's are undetectable except by very sensitive measurements. In the known cases, like the present one, they seem to become systematically nonexistent (the C_p jump, ΔC_p , disappears) as the modulator (or network breaking component) is removed. The significance of vanishing T_g to theory of the T_g is currently at the center of debate.”

In previous manuscript we follow Angell's terminology and refer to Al-BHET, Al-MTBT, Al-BHET all as glasses, as the T_g also increase with decreased modulator amount (Fig 4). **In the revised manuscript, we now only use the term “glass” for materials that have well-defined T_g confirmed by DSC and DMA to avoid controversy.**

Fig. R2. DSC and DMA of Al-BHET and Al-BHET-TPTO, which is now fig. 3b, 3c, 5b, 5c in the revised main text.

We have now found that the as-synthesized Al-BHET, Al-BHET-TPTO all have T_g , and the Al-MTBT does not (main text figure 3b, 3c, 5b, 5c, summarized here in fig. R2). The T_g was both found by DSC measurement and DMA measurement, indicating the configurational motions that are unfrozen at T_g are associated with overall motion of the framework. Thus, we would refer to the Al-BHET series as glasses and Al-MTBT as “vitrified monolith”. Moreover, even with other modulators that do not show T_g by itself, the Al-BHET series also show T_g , indicating the configurational motion at T_g is not associated with TPTO (fig. R3).

Fig. R3. (a) a list of other modulators used for synthesizing Al-BHET-modulator (b) even for modulators that do not show T_g , the Al-BHET-modulator show T_g

Notably, in the previous manuscript, the T_g of Al-BHET-TPTO was measured after additional vacuum heating at 120 °C after synthesis, which is a precaution procedure to remove solvent. However, we now found that such additional process affects the T_g (the as-synthesized Al-BHET-TPTO glass show T_g of 6 °C and the vacuum heated sample have T_g around 20 °C). Considering that in the actual activation process, the as-synthesized glass is directly subject to solvent exchange, we now measured the T_g of as-synthesized glasses without such additional vacuum heating to keep consistency.

No glass transition was found in the DSC and DMA for the activated Al-BHET and Al-BHET-TPTO, indicating the configurational motion of the framework at T_g involves both the framework and the guest molecule in the pore, which means the modulators in the pore have plasticizing effects. On the molecular level, such plasticizing effect can be easily comprehended in analogy to the modulators in MOF chemistry: the modulator in the pore could dynamically exchange with the linker and facilitate the configurational motion of the framework necessary for rheological behavior (ref. 15). It is important to note that the activation process only involves solvent exchange, supercritical CO₂ drying and evacuation, all at room temperature, which is a very mild procedure adopted directly from the activation of delicate MOFs. This mild process only removes guest molecules in the pore and does not affect the coordinative backbone, which can also be shown by the solid-state NMR and pair distribution function of the as-synthesized and activated material (**figs 4d, 5f, S39, S54**). Thus, the activated glass that show high porosity can be viewed as the coordinative network backbone of a glassy material with well-defined glass transition.

From the reasoning above, we could draw conclusion on the glassy nature of the Al-BHET series. The as-synthesized aluminum alkoxides glass has coordinatively linked porous framework filled with guest molecules, which are modulators that have hydroxy functional group. These guest molecules play the role of plasticizer by coordinatively perturbing the framework, which give rise to well-defined glass transition. The modulators also stabilize the pores during glass synthesis by occupying the pore volume, which can be removed in a mild activation process. The pore volume can then be measured by gas uptake, which shows the high porosity of the aluminum alkoxide glasses.

2. In the “High-porosity MOF@glass composites” section (lines 222–229), the term "glass ceramics" is inaccurately used. The term "composites" seems to be more suitable in this case.

We agree that the term “glass ceramics” is not suitable for this case, and we have changed the term to “composites”. Considering the recommendation from reviewer 4&5, we have now moved this session to Supporting information.

3. EXAFS and XAS discussions (lines 214–215) are unclear. How can the presence of AIOC be confirmed by comparing EXAFS and XAS with alumina? In addition, I recommend adding actual EXAFS and XAS data for alumina to Figures 5C, S6, and S7. Is Figure 5C identical to Figure S6?

We are sorry for the confusion in this section. The XAS measurement is used as a corroborate evidence to support the PDF and compositional analysis to show that there are AIOCs in Al-BHET

glasses. We use the alumina in the comparison of near edge feature because it is a common concern that aluminum alkoxide would hydrolyze to form alumina during high temperature synthesis.

Fig. R4. (a) XAS near edge feature for Al-BHET, AIOC-12 and alumina, which is the figure 4c in the main text; (b) the R-space pattern generated from the EXAFS data of Al-BHET and AIOC-12, which is a corroborating evidence for the presence of AIOC (now fig. S25 in the revised manuscript)

The XAS data for alumina and a well-defined AIOC-12 (a stable AIOC that is similar the AIOC-41 in terms of aluminum chemical environment that is suitable for XAS study) is measured (fig. R4), and the comparison shows that the near edge feature of Al-BHET is very much similar to the AIOC and distinctively different from alumina. For EXAFS, due to the Al K edge is in soft X-ray regime, and the aluminum alkoxide is very insulating, thus the data quality is poor even extensive integration time is used. Consequently, it is not feasible to quantitatively fit the EXAFS data. However, from the R space pattern, the EXAFS data also qualitatively support that the Al-O bond length in Al-BHET is consistent with AIOC.

4. The thermal stability of all samples should be discussed.

The stability of glasses and monoliths is shown by TGA and discussed in the main text, the thermal stability for the activated glasses and monoliths are all above 150 °C. The thermal stability of the as-synthesized glass near the glass transition temperature is confirmed by the reversibility of the DSC curves.

Fig. R5. TGA and three-round DSC scan for Al-BHET (a) and Al-BHET-TPTO glasses (b), these data are now in the SI of the revised manuscript (S17, 18, 33, 34). The TGA of Al-MTBT monolith is now put in fig. S7.

5. Figure 6b requires the isotherm of pristine UiO-66, and the mass loading of UiO-66 should be described in the manuscript.

The glass-crystal composite has been removed to SI, and the mass loading is described in the supporting information : “*The UiO-66 is added to the above-mentioned Al-BHET synthesis (200 μ L Al-BHET solution per 8 mg UiO-66).*”, which gives the mass loading of UiO-66 of around 50%.

6. I suggest using a linear scale for the x axis in all pair-distribution function data.

The scale of the PDF has been revised to linear scale in the figs. 4b, 5f and S13, S42.

Overall, the quality of the figures, terminology, discussions, figure captions, and data interpretations are unclear. Throughout the manuscript, several explanations and discussions are mentioned without appropriate citation or direct evidence. A careful revision and additional experiments are necessary.

We thank the reviewer again for raising questions and concerns that help improve this paper, we have now substantially revised the manuscript and we hope the new version would be satisfactory.

Reviewer #2 (Remarks to the Author):

In this manuscript, Prof. Zhao and his colleague reported a type of new glass forming metal organic glass materials, fabricated through templated removal of the organic solvent. The method is versatile and the resultant materials can preserve a large amount of porosity. This materials is one of the more porous glass reported in the literature and it would find good use in many applications. The materials characterisation is clearly in depth - with advanced characterisation techniques including x-ray absorption spectroscopy and total scattering, the atomic pair distribution and fine structure can be retrieved, which further shed lights onto the arrangement of the metal-organic hybrid materials. The authors have also demonstrated the capability of compositing with other functional materials like UiO66. Overall, I think this paper is of high quality and can be published in nature communication after some minor revision.

We really thank the reviewer for their support on this paper.

1. The author states the materials features good glass forming capability - would heating the liquid MOF to a higher temperature and then quenching back to RT lead to any re-crystallisation?

The issue of glass forming ability for bottom-up synthesized coordinative networks are quite different from that of ZIFs, as these networks do not really have a crystalline state. Specifically, for Al-BHET and Al-MTBT, they also do not have crystalline counterpart and we have not observed recrystallization due to annealing. However, crystallization of the linker or by-products can be observed in certain cases. For example, as MTBT has low solubility, it would precipitate from the liquid phase during evaporation when the solvent mixture is less ideal, or the evaporation temperature is not well-controlled. Comparably, the synthesis of Al-BHET is much more robust. From our experience, for the bottom-up synthesis of glassy coordinative framework, the process that hinders vitrification is usually not the recrystallization of the coordinative framework, but the phase separation between linker or metal node from the homogenous liquid phase accompanied by the crystallization of one constituents of the framework. Thus, "glass forming ability" in this context refer to the tendency of the system to maintain a homogenous liquid without one of the constituents undergoing side-reactions or crashing out of the liquid state.

2. Is it possible to provide some data on the mechanical properties of these materials? Given the T_g is around RT range, would that means the materials behave more like a gel under ambient?

The modulus of the as-synthesized glasses is measured by rheology and nano-indentation (**figs. S9, S21, S22, S37, S52**) which shows these glasses remain in solid form at room temperature with slight elasticity. Notably the mechanical properties of these samples are at the limit of rheological measurement (very close to powder and does not flow well-enough). For the nano-indentation measurement, the sample are brittle and also difficult to get quantitatively reliable modulus. Nevertheless, we think it is safe to say these glasses are more like polymer powder than gel or liquids at room temperature. From a fundamental understanding, we speculate that the mechanical properties of the aluminum alkoxides can be better understood with the consideration of the following two aspects:

- (1) The temperature dependence of mechanical properties for glassy material is often described by Angell plot (*Nature* 2001, 410, 663–667; *J. Non-Cryst. Solids*, 1991, 131-133, 13–31). Empirically, materials with covalently linked networks (e.g. SiO₂) are known to be “strong” glasses, where the material’s modulus change relatively slowly with temperature. In the perspective, the aluminum alkoxide linked with charged Al-O bonds are also strong glasses that remain in solid form even at temperature above T_g .
- (2) We envision that the aluminum alkoxide glasses would be used in their activated form. With the modulator in the pore removed, which have plasticizing effect, the glassy framework does not show T_g in DSC or DMA, thus are truly in solid form. As a matter of fact, the main limitation of these glasses is their brittleness, which we are still working to resolve.

Notably, in the previous manuscript, the T_g of as synthesized Al-BHET-TPTO was measured after additional vacuum heating at 120 °C, which is not applied during the sample activation. In the subsequent study, we found that such heating affects the T_g . Thus, to keep consistency, we now measured the T_g of all as-synthesized glasses without additional vacuum annealing.

3. The large hysteresis effect as evidenced by the gas isothermo need further discussion and clarification, as this may affect the practical use of the highly porous materials.

We agree that the hysteresis of these isotherms would affect the practical uses of these materials. As evidenced from the data presented in the paper, Al-BHET has large hysteresis in CO₂ uptake, and the Al-MTBT and Al-BHET-TPTO has smaller hysteresis. This is likely due to the small pore volume of Al-BHET that hampers desorption, which also lead to long equilibrium time in the measurement. The hysteresis could also partially originate from the network flexibility, which is a common cause for hysteresis. As the BHET linker is flexible and glassy network is a kinetic product, it is understandable that the framework could have certain structural change in the gas sorption process. As for methanol uptake, the hysteresis would be likely due to the chemisorption of methanol to the framework. We would certainly investigate these aspects in the future, however, we prefer not to discuss these aspects in this main text due to the complexity and uncertainty related to this matter to avoid confusion or misinformation.

Reviewer #3 (Remarks to the Author):

This manuscript proposes a strategy for synthesizing coordination network glasses with high internal surface. The authors have synthesized two kinds of metal-organic glasses using a solvent evaporation approach. To synthesize Al-BHET and Al-MTBT glasses, they dissolved aluminum sec-butoxide together with BHET or MTBT into glacial acetic acid/butanol/ethanol or ethylene glycol methyl ether/THF/ethanol solutions, respectively. The resultant solutions were heated at 125 °C to evaporate the solvent solutions and obtain glassy samples which demonstrate gas adsorption capability and low glass transition temperatures (below 0 °C). They also incorporated TPTO into the Al-BHET and Al-MTBT glass networks as a modulator which can be removed by alcohol exchange and finally increase the porosity of glass. The prepared glasses possess high porosity. It is an interesting work for the structure control and porosity design of coordination glasses. Yet, the local structure and chemical compositions of the prepared glasses is not clear enough. Numerous issues need to be addressed. Due to the low T_g values (below 0 degree), the application of these glasses at ambient conditions will be limited. This manuscript requires major revisions before potentially being published in Nature Communications. My specific comments and suggestions for improving the manuscript are as follows:

We thank the reviewer for appreciating the novelty of this work, and we have now significantly revised this manuscript to increase clarity and readability to a more general audience.

1. The authors used metal:organic linker molar ratios of 1:1, 1:1.2, and 4.15:1 for the preparation of the Al-BHET, Al-BHET-TPTO, and Al-MTBT glasses, while there is no experimental procedure for the preparation of Al-MTBT-TPTO glass. Why did the authors use these molar ratios? What is the chemical composition of each glass? How does this metal:organic molar ratio influence the glass properties in terms of porosity and glass transition?

The molar ratios are initially chosen according to the number of OH groups in the linker and then maintained throughout this study for consistency. The chemical composition of each glass (activated) is shown in table S4, and the composition of Al-MTBT and Al-BHET-TPTO are determined by digested NMR and elemental analysis to be $\text{Al}_8\text{O}_2(\text{OH})_{4.9}(\text{MTBT})(\text{EGME})_{2.7}(\text{CH}_3\text{COO})_{8.4}$ and $\text{Al}_8\text{O}_2(\text{OH})_{8.8}(\text{BHET})_{2.4}(\text{CH}_3\text{COO})_{6.2}(\text{TPTO})_{0.2}$, respectively.

Table S4. Summary of elemental analysis

Sample Name	C/%	H/%	O/%	Al/%
Al-BHET	43.1	4.9	42.8	10.3
Al-BHET-TPTO	36.7	4.7	43.7	11.9
Al-MTBT	43.2	5.6	39.8	12.4

Notably, slight variation in metal-to-linker ratio (e.g. changing from 1:1 to 1:1.5) in the synthesis would not fundamentally affect the framework formation, as the excess linker would stay in the pore as guest molecules and later be removed in solvent exchange. For the same reason, the effect of changing the metal-to-linker ratio in the initial synthesis on material porosity is also complex and not monotonic. Moreover, as Al-BHET generally have low porosity, it would not be fruitful to try improving its porosity by varying the linker-to-metal ratio, and a more promising way to increase

its porosity is introducing TPTO, as we have shown in the main text. As for glass transition, it was shown in the previous manuscript (fig. R5, previous fig. 4) that with increased OH: Al ratio by adding TPTO, the T_g would be lowered. And it would be natural that excess amount of linker would have the same effect of excess TPTO, as most of these linkers would not be bonded to the AIOCs anyway. On the other hand, lowering the linker amount below the current value would lead to uncoordinated AIOCs that are left as impurities in the network. However, such discussion in the main text would lead to substantial confusion and it has little connection with the main findings of the paper, which is achieving high porosity for coordinative glasses, thus we now remove figure 4 from the previous manuscript and also would not discuss these detailed chemistry in the main text.

Fig. R5. Increased OH: Al ratio lead to lower T_g

2. The authors used solvent exchange method to remove TPTO by THF and acetone. Based on NMR and FTIR data, they claimed that only a very small amount of TPTO is left in the activated glass compared to the amount initially added. However, the FTIR data confirm that large amount of TPTO is left in the glass and BHET who dissolved during solvent exchange. The intensity of the resonance peak at 1100 cm^{-1} (TPTO) increases, while, the intensity of the resonance peaks of BHET decreases after solvent exchange as shown in Figure S19.

It is fundamentally inappropriate to allocate an IR absorption in this range (near 1000 cm^{-1}) to certain species in such complex materials. From both the basics of IR spectroscopy and the experimental data, it can be shown that both BHET and TPTO have vibrations that give absorption in this range. Thus, the digested NMR is the most appropriate quantitative method to analyze the ratio between BHET and TPTO.

3. The authors mentioned that their glasses are the most porous glassy materials with high surface areas up to 500 m^2/g . This is incorrect, there is another metal-organic glass based on titanium clusters has been reported with surface areas of 923 m^2/g which is much higher than that reported in this manuscript (Xu, W. et al. High-Porosity Metal-Organic Framework Glasses. *Angew. Chem. Int. Ed.* 62, 278 e202300003, 2023).

The titanium carboxylate monolith shown in this paper have shown no evidence that it is a glassy material (e.g. no DSC curve or DMA measurement) and thus cannot be categorized as glasses.

4. As shown in Figure 1, the authors mentioned that the aluminum sec-butoxide reacts with BHET/TPTO or MTBT/TPTO during solvent evaporation at 125 $^{\circ}\text{C}$, forming coordination bonds

such that aluminum cluster bridges two BHET or MTBT. Could you provide evidence for this assumption?

The fig. 1e is a schematic showing the general idea of pore-templating, and the dash line in this figure in a common way in reticular chemistry to show that other connections to certain chemical entity are omitted for clarity in a schematic drawing. We have now revised the figure caption to clarify this: “*The porosity of aluminum alkoxide glasses can be substantially increased with pore-templating modulator, which is shown in a schematic (e)*”.

5. The simulated and experimental PDF data show the short-range atom-atom correlations in the aluminum clusters. As is known, ZIF-glasses show a high degree of short-range disordered structure due to the weak coordination bonds and large ligands (see Madsen et al, Science 367 (2020) 1473-1476). The authors should describe the short-range structure in their glasses in terms of disorder or order. From Fig. 1, it seems that the medium range structure of your glasses is of high degree of order. The authors should comment on this.

The short-range features in the PDF showing the Al-O, C-O, Al...Al (Al-O-Al) and Al...O (Al-O-Al-O) correlation originate from the atomic order inside the AIOC. Thus the aluminum alkoxide glasses would have short-range order as long as the AIOCs are intact, regardless of how they are connected by the linker to form the network. In comparison, the metal node of ZIFs only have one atom, the Zn atom, and the imidazole linker connecting the adjacent Zn atoms in ZIFs are rigid and small, thus the short-range ordering showing Zn-N and Zn...Zn correlation already reflects how the coordinative networks are connected. Thus, the length scale of ZIF and Al-BHET is very different, where the short-range ordering of ZIFs provide information of the network connectivity whereas the short-range ordering of Al-BHET does not. Consequently, the ordered short-range structure for Al-BHET does not convey the same type of information as the ZIFs. As for the medium range ordering of Al-BHET, which would in principle show the correlation between the AIOC metal nodes, the PDF in this range cannot be directly interpreted due to the flexibility of BHET linker. To analyze the degree of connectivity of the Al-BHET from PDF, extensive modeling and fitting is required, which we would like to pursue in the future.

6. Furthermore, the FTIR data strongly suggest that the coordination bonds are not formed in the prepared glasses. The resonance peak of OH group in the BHET/MTPT linkers at 3500 cm⁻¹ is clearly observed in all the spectra of the prepared glasses as shown in Figure S4,S13,S19,S25. This means that the BHET/MTPT linkers have not deprotonated during solvent evaporation. The authors also should provide the FTIR spectrum of the aluminum sec-butoxide for comparison to clearly see what happened between aluminum clusters and BHET/MTPT linkers during solvent evaporation.

The IR spectrum of aluminum sec-butoxide is measured and compared with aluminum alkoxide glasses (figure S10, S23, S38, summarized in fig. R6a). The formation of coordination bonds can be analyzed as follows:

(1) From the IR, the activated glass has very low IR absorption at 3500 cm⁻¹, which can be allocated to adsorbed water, bridging hydroxyl group in the AIOC and partially connected linker at the framework defect. Notably, crystalline AIOCs also show IR absorption at this range due to bridging

hydroxyl groups and adsorbed water (fig. R6b, adopted from ref. 17). Considering the substantial amount of BHET in the activated glass and the weak absorption at 3500 cm^{-1} compared to that of BHET linker, the majority of BHET are certainly deprotonated. As for the as-synthesized aluminum alkoxide glasses, the presence of 3500 cm^{-1} peak may come from the alcohol modulators in the pore, including residue solvents, TPTO modulator and unreacted linkers.

Fig. S37 FT-IR spectra of AIOC-2, AIOC-6, AIOC-8 and AIOC-11.

Fig. R6. (a) IR spectra of aluminum alkoxides; (b) IR spectra of AIOCs, adopted from ref. 17.

(2) The deprotonation of BHET can also be deduced from the activation process, which involves extensive washing with THF and THF/ethanol mixture. This process would certainly dissolve BHET that is not coordinatively bonded to aluminum.

(3) The formation of AIOCs, regardless of their exact structures, almost always require the coordination of aluminum with deprotonated alcohol. In Al-BHET, the only alcohol found in digested NMR is the BHET, and the presence of AIOC is confirmed by PDF and elemental analysis, thus from a chemical perspective the BHET does coordinate to AIOC without reasonable doubt.

7. In my opinion and based on the performed structural characterizations for the prepared glasses, the dissolution of aluminium alkoxide and BHET/MTPT into alcohols (e.g., THF, ethanol, butanol) followed by thermal treatment at $125\text{ }^{\circ}\text{C}$, results in the solvent evaporation and the formation of 3D hydrogen bonded molecular glass network. To some extent, this glass has some similar features to the previously reported metal inorganic-organic hybrid glass with supramolecular network (Ali, et al, Angew. Chem. Int. Ed. 2023, 62, e202218094). In this glass, aluminum clusters, solvent, and BHET/MTPT/TPTO are connected with each other through hydrogen bonds. The organic linkers may coordinate with aluminum clusters forming large molecules but the coordination bonds do not form in this glass as they mentioned in their discussions. The coordination of organic linkers with aluminum clusters should be examined carefully. Therefore, the structural related discussions should be revised. The origin of the porosity in the aluminium alkoxide glass is not clear. Is it due to the structural property or is it induced during solvent exchange process which results in the dissolution of glass constituents and form pores? The authors should examine the porosity and the

composition of the glass samples especially before and after solvent exchange and activation process. This could help them to understand the origin of the porosity for their glasses.

We encourage the reviewers to refer to general literatures on MOF chemistry to get a better picture of the chemistry presented in this manuscript (e.g. *Science*, 2013, 341, 1230444). To clarify the difference between the aluminum alkoxide glass and the supramolecular network reported in the *Angew. Chem. Int. Ed.* 2023, 62, e202218094, it is important to understand the connectivity of the network from a basic chemical perspective. In the synthesis of MIOC in the Angew paper, stoichiometric zinc nitrate and imidazole are dissolved in ethanol and evaporated to give the glassy product. As nitrate is not volatile nor basic, and there is no additional base, the imidazole linker remains completely protonated, which is clearly stated in the paper (fig. R7, also if the imidazole were to be deprotonated, then it would produce nitric acid, which contradicts fundamental concept of acid base chemistry). Thus, each imidazole linker can only coordinate with one zinc ion without forming the charged Zn-N bond in ZIF, and the material is essentially a supramolecular network of zinc coordination complex. If the metal source were to be zinc hydroxide instead of zinc nitrate, then the hydroxide base would be able to deprotonate the imidazole linker to give a coordinatively linked network. This also echoes the very common practice of MOF chemistry where a base is always needed to deprotonate the linker to give MOFs (i.e. DMF can give dimethylamine when heated; acetate can facilitate MOF growth more than nitrate due to higher basicity; ZIF synthesis with zinc nitrate in water/alcohols requires large excess of imidazole to maintain basic condition, *NPG Asia Mater.* 2020, 12, 58; *Chem. Eur. J.* 2024, 30, e202304256; *Chem. Mater.* 2018, 30, 10, 3467–3473).

For Al-BHET, the aluminum tri-butoxide is a strong base that is adequate to deprotonate BHET, and the resulting material would naturally be a coordinative network. Alternatively, the reaction of aluminum tri-butoxide and BHET can also be viewed as a coordinative exchange reaction, where the butanol ligand is exchanged with BHET, which has the same hydroxyl functional group. As the BHET is not volatile but butanol is continuously being evaporated from the synthetic mixture, then the equilibrium would naturally shift in favor of BHET coordinating with aluminum. Conversely, if aluminum nitrate is added instead of aluminum butoxide, then evaporation would give product similar to this Angew paper, which is supramolecular mixture. This is in analogy to the facile preparation of MOFs with acetate salt: zinc acetate can react immediately in room temperature with terephthalic acid to give MOF-5 (*Mater. Chem. Phys.* 2011, 131, 358–361), as acetate is basic enough to deprotonated benzoic acid; or it can also be viewed as the carboxylic group of terephthalic acid exchange with the acetate for zinc coordination. However, MOF-5 cannot form if zinc nitrate is mixed with terephthalic acid, unless extra base is added, or in-situ formed by DMF decomposition.

Sample	Measured Him/(Hbim+Him)	Composition (wt %)
ZIF-7-III crystal	0	Zn ₂ (bim) ₄
Undried MIOC glass	0	88 Zn(NO ₃) ₂ (Hbim) ₂ ·12 Ethanol
MIOC glass	0	96.5 Zn(NO ₃) ₂ (Hbim) ₂ ·3.5 Ethanol
Brown MIOC glass	0.03	100 Zn(NO ₃) ₂ (Hbim) _{1.54} (Him) _{0.06}

Fig. R7. The chemical structure of the MIOC in *Angew. Chem. Int. Ed.* 2023, 62, e202218094

The presence of coordinative network can also be deduced from a simple fact that Al-BHET is not soluble in common solvents such as DMSO, THF and alcohols. As BHET is highly soluble, and these polar solvents can readily disrupt hydrogen bonding, any BHET hydrogen bonded network that is not coordinatively bonded would be dissolved. However, in the activation process, the solvent wash was carried out by solvent exchange three times a day for up to a month, and the resulting Al-BHET-TPTO is not dissolved and still show significant presence of BHET, which shows the BHET are bonded to aluminum. Actually, extensive solvent wash has been a long-established method to distinguish between coordinative framework and oligomers (*Nature*, 1995, 378, 703-706).

In addition, the presence of gas accessible porosity is also by itself an evidence for coordinative network in Al-BHET. Although hydrogen bonded materials can have porosity, their building blocks need to be rigid to maintain pores (e.g. *Chem* 2022, 8, 2114–2135). For flexible molecules such as BHET and TPTO, hydrogen bonding would certainly not be able to give gas accessible porosity. The MIOC in the *Angew. Chem. Int. Ed.* 2023, 62, e202218094 paper also show no porosity.

The composition of the Al-BHET-TPTO sample after solvent exchange and activation is studied by digest NMR, which shows the TPTO template is removed in this process. Such process is similar to removing DMF from the MOF-5 to give pores (*Nature*, 1999, 402, 276-279). Without activation, there would certainly be no gas accessible pores.

8. In lines 57, 80 and 88, the authors claim that the viscosity of clear and colorless solution increase and then the vitrification occurs. However, the clarity of this process remains obscured solely by the authors' statements. Before activation, the product likely encompasses various components such as solvent molecules, by-products, and unreacted materials. How to prove the sample turns to glass rather than the very sticky solution? What is the product looks like? Is it hard or soft? Given the imperceptible glass transition of Al-BHET and Al-MTBT glass, it prompts consideration of whether the products synthesized are well-defined glass or only amorphous materials.

The activation process would indeed remove various guest molecules in the pore. The glassy nature of the as-synthesized Al-BHET and Al-BHET-TPTO glasses is now clearly shown from the DSC and DMA measurement (**figs. 3b, 3c, 5b, 5c**). The T_g of the as-synthesized material is below room temperature so they are elastic solids, which is shown in the rheological and mechanical measurements (**figs. S9, S21, S22, S37, S52**). Al-MTBT does not have well-defined T_g so we would refer it as “vitrified monolith”.

In a broader term, as the viscosity of a liquid would increase exponentially when cooled towards T_g , glasses are actually “very sticky liquid”. Actually, one definition of glass transition temperature is the temperature where super-cooled liquids reach certain viscosity (10^{12} poise). The static structure of glasses and the corresponding super-cooled liquids are also difficult to distinguish (*Nature* 2001, 410, 663–667; *J. Non-Cryst. Solids*, 1991, 131-133, 13–31; *Chem. Rev.* 2019, 119, 7848-7939).

9. In lines 96 and 98, as well as in Figure S2, the authors mention the possibility of crystallization occurring during the preparation process. The SEM images with TPTO modulator also reveals heterogeneous shapes. Are these morphologies indicative of crystallization, or do they signify the

presence of impurities? Further clarification is needed for the crystallization that produced during the preparation process.

The SEM shows fractured glasses and wrinkles caused by surface tension of liquids when vitrification took place. These features actually show the rheological behavior and are not indication of impurities. Crystallization can be readily avoided in the optimal synthetic condition presented in the paper. In less ideal synthetic condition such as uncontrolled evaporation, MTBT itself could precipitate out due to low solubility. Exposure to moisture in the initial stage of the reaction could also produce Al-containing unknown species due to uncontrolled hydrolysis. Notably, the presence of byproducts is common for the preparation of discrete AIOCs, and the composition and controlled synthesis of these byproducts is beyond the scope of this work. Currently we would report the conditions that do not produce undesirable products.

10. In line 109 and Figure S3, the author attempted to demonstrate the porosity of the product through a methanol adsorption test. However, it is worth considering that the glass may harbor numerous reaction points with methanol, leading to the connection of gas molecular and glass structure rather than being absorbed within the pore of glass structure.

We agree with the reviewer on this point. Thus, methanol uptake is only used as a corroborating evidence for porosity, and the pore volume is calculated from the adsorption date of N₂ and CO₂. As we did not observe a steep uptake of methanol at very low pressure, we would speculate the methanol to have both chemical and physical adsorption.

11. In the section discussing "chemical space and glass transition," the author used DSC to analysis variation in glass transition with different TPTO ratios. However, according to the supporting information, it remains unclear whether the test samples underwent an activation process. If the response is negative, the presence of residual TPTO could potentially impact the results, leading to a gradual attenuation of the glass transition signal. This scenario poses a challenge in conclusively establishing the glass transition nature of the samples.

We agree that this figure could be confusing, and it is now removed. These samples are not activated and studied solely for the purpose of understanding the effect of network connectivity on glass transition (See response to the first question to the first reviewer) and have no connection with the samples studied for porosity. We have now removed this figure and discuss the glassy nature of the materials with DSC and DMA.

12. In the left panel of Figure 4, please indicate where T_g is and how to determine the T_g values. Please describe the conditions for the DSC measurement, e.g., upscan and downscan rates, atmosphere, pressure. The standard T_g values should be measured under the standard conditions (Zheng et al. Chem. Rev. 119 (2019) 7848-7939). Fig. 4: Please label a) and b) inside the figure.

The T_g in the DSC curve is determined with the routine inflection method that is built-in for the TA data processing software, the measurement condition is also routine (10 K/min) unless specified

otherwise. The measurement is in 1 atmosphere with N₂ purging the cell, which is also routine and specified in the Supporting Information.

13. Since the T_g values are below 0 degree, what kind of applications do you expect?

The T_g of these glasses would depend on the guest molecules in the pore that have plasticizing effect (TPTO, residue butanol, etc), which facilitate framework configurational motion. With these guests removed by solvent exchange, and the activated glass no longer show measurable T_g or rheological behavior before thermal decomposition. Thus, in practical applications, the glasses can first be processed in the as-synthesized form into desirable morphology and then activated by solvent washing and evacuation, then the activated glass would combine high porosity and processability. As for applications, the porous, processible and fluorescent glasses can be used for gas permeable/responsive coatings for gas sensing devices. It can also be used to host porous crystalline MOFs to produce composite porous materials with well-defined shape and morphology, which would be superior to compressive shaping of crystalline MOFs that often compromise their porosity and crystallinity.

14. Figures S4, S19 and S25 show the FTIR spectra of Al-BHET, Al-BHET-TPTO and Al-BHET-TPTO-2. The syn samples exhibit a good alignment, whereas the ads samples do not match as closely with each other. What makes the difference? Is it possible that the solvent exchange process damages the sample's structure? Alternatively, could varying TPTO ratios lead to structural changes in the glass after activation?

The FTIR spectra of as-synthesized sample resembles the BHET linker more because there are free BHET molecules trapped in the pore. During activation, these guests are removed, thus giving changes to the IR spectra. This is a physical process and would not change the structure. As for the relationship between the as-synthesized and activated glass, it is important to note that the activation process only involves solvent exchange, supercritical CO₂ drying and evacuation, all at room temperature, which is very mild procedure adopted directly from the activation of delicate MOFs. This mild process only removes guest molecules in the pore and does not affect the coordinative backbone, which can also be shown by the solid-state NMR and pair distribution function of the as-synthesized and activated material ((**figs 4d, 5f, S39, S54**)). It is worth noting that FTIR essentially reflect local chemical structures, and below 1600 cm⁻¹, the vibration of linker, AIOC and various bonds overlap in a complex way, thus it would not be fruitful to speculate on the overall structure of the framework with FITR, which mostly concerns the connectivity and collapse of pores. The FTIR shows the deprotonation of the linker, and the perseveration of linker functional groups such as the carbonyl group, as a corroborate evidence for the chemical composition analysis.

Varying the amount of TPTO would certainly lead to different structure, as it influences the structure in several aspects that are convoluted in complexed manners. Such complexity would be subjected to future research.

15. In Figure S24, there is a noticeable alteration in the PDF diagram following various solvent

exchange processes. The question arises: is this change attributable to an influence of different solvent on the glass structure, or does it suggest the penetration of solvent into the glass structure?

The sample measured for PDF are dried before measurement, and the prominent features in the short-range are persevered during the activation (fig. 5). The differences for relative peak intensities are caused by removing the modulators in the pore changes the relative ratio of different species in the material (*i.e.* BHET, TPTO, AIOC).

16. The section on "High-porosity MOF@glass composites" appears brief and simplistically described. SAED and EDS analyses could provide a more comprehensive understanding of the composite structure. Elaboration on how these components interconnect and an exploration of the impact of the glass's chemical and physical properties on UiO-66 would contribute to future's work.

We thank the reviewer for the suggestion. The EDS of the composite is performed to corroborate the presence of UiO-66 (**figs. S27-S28**). Nevertheless, we agree that the spatial distribution of the MOF nanocrystals is important for applications, which we would pursue. However, considering the recommendations from reviewers 4&5, we would move this section to SI and focus on the glass itself in the main text.

17. There are some grammars and writing problems (for instance, in line 92, "solvent mixture" should be amended to "mixture for solvent."). Please carefully review and polish the entire text.

We have revised this statement.

18. The figures and their respective captions need to be improved to facilitate understanding for the authors.

We have revised the captions to be more informative for the general audience.

Reviewer #4 (Remarks to the Author):

In this work, Zhang et al. developed a new series of aluminum alkoxide glasses through coordinatively competitive solvents evaporation. Meanwhile, they also provide a generalizable method to balance the porosity and glass forming ability of MOF glasses by using modulator templating approach. This strategy sheds the light for forming some porous MOF glasses with flexible ligands. But it's important to point out that this strategy needs more validation. For example, during the process of washing the sample to remove template molecules, will the ligand BHET be also washed away? How much TPTO remains in the final porous sample? This requires NMR and GC characterization of the sample after acid hydrolysis. DSC results in Fig4 imply that the TPTO content in the sample may be large. In other words, the porous skeleton in the Al-BHET-TPTO sample is actually constructed from BHET and TPTO. This is actually very contradictory to the porosity formation mechanism proposed by the author (Fig1e). By the way, some similar synthesis strategies have been already reported (Ref 15, 16), which reduces the novelty and importance of this work to some extent.

The activation process involves solvent washing, which certainly also remove unreacted BHET along with TPTO. The activated Al-BHET-TPTO is studied by NMR, where the sample is prepared by digestion under basic condition. Notably the aluminum alkoxide clusters are highly stable towards acid, and the low volatility of BHET and TPTO prevent the usage of GC for quantification. Digested NMR shows the molar ratio BHET:TPTO=11.8 for the activated glass, which is much higher compared to the ratio added in the synthesis (BHET:TPTO=4.35), indicating most of the TPTO is removed in the activated glass.

In figure 4, the large excess of TPTO is meant to show the plasticizing effect of monodentate alcohol, which has no relationship with the samples studied for porosity (please see the response to question 1 for the first reviewer). In the current paper, now that the Al-BHET and Al-BHET-TPTO have well-defined glass transition (**figs. 3b, 3c, 5b, 5c**), we removed figure 4 to avoid misunderstandings.

As is stated above, the bottom-up synthesis of titanium phenolates and carboxylates that show monolithic features have been reported. However, none of these materials simultaneously show well-defined glass transition and gas accessible porosity due to the trade-off on linker rigidity. In this report, we found that by introducing a bulky modulator as template, flexible linkers that give high glass forming ability can also produce highly porous frameworks. In addition, the aluminum alkoxides can have high transparency, low density and mild synthetic conditions, whereas the titanium phenolate requires high temperature (160 °C) evaporation of toxic cresol solvent and have dark red color due to the titanium-phenol bonds. Thus, we believe the current paper represent an important advance for metal-organic network-forming glasses.

1. In Figure 1d, Al-BHET glass seems very optically transparent, but how about the transparency of Al-BHET-TPTO glass?

The Al-BHET-TPTO glass is also transparent, which is now shown in fig. 5a.

2. One of the indicators to determine the glass formation is whether the material is stable within the glass transition temperature (T_g) range. Could you provide more detailed thermal stability information of your glasses?

The thermal stability of the activated glasses is shown by TGA. For the as-synthesized glass, as the T_g is too low for TGA measurement, its thermal stability near T_g can be shown by repetitive DSC measurement, where the Al-BHET and Al-BHET-TPTO all show reversible DSC curve indicating the glasses are stable in this temperature regime (fig. R5, see also the response to reviewer 1).

Fig. R5. (this figure is the same as the one in response to reviewer 1): TGA and three-round DSC scan for Al-BHET (a) and Al-BHET-TPTO glasses (b), these data are now in the SI of the revised manuscript (S17, 18, 33, 34). The TGA of Al-MTBT monolith is now put in fig. S7.

3. Structural analysis is very important in this work. I think the data quality of the XAFS results is very poor (Figure 5c, Figure S6 and Figure S8), which makes the current analysis results untrustworthy. If data quality can be improved, authors need to provide necessary data fitting.

As the Al-BHET is highly insulating and the Al K-edge is in soft X-ray regime, the signal-to-noise ratio in the EXAFs regime is indeed quite poor, which is not uncommon for Al EXAFS and thus prevent a quantitative fitting for the aluminum coordination environment. However, we use the XAFS only as a corroborating evidence that support the presence of AIOC: (1) the near edge features of Al-BHET resembles that of AIOC-12 and are different from alumina; (2) the R space plot for Al-BHET and AIOC-12 qualitatively shows that the Al-O bonds in these two materials have similar length. Thus we believe the XAFS data indeed serve as a supporting evidence that is consistent with the presence of AIOC incorporating alcohol and carboxylic acid ligand in Al-BHET.

Fig. R4. (this figure is the same as the one in response to reviewer 1): (a) XAS near edge feature for Al-BHET, AIOC-12 and alumina, which is the figure 4c in the main text; (b) the R-space pattern

generated from the EXAFS data of Al-BHET and AIOC-12, which is a corroborating evidence for the presence of AIOC (now fig. S25 in the revised manuscript).

4. For better comparison, it is better to provide the results of AIOC in both Figure 5b and Figure 5c.

We obtained AIOC-12 as the reference sample in XAS due to its high stability, synthetic viability, and structural similarity with AIOC-41 concerning the local structure and valency of aluminum. The AIOC-41 is unstable at ambient moisture and can degrade during sample transferring to synchrotron beamline.

5. Considering the complexity, is it possible for the author to detailly explain the process of obtaining simulation results of pair-distribution function again?

The simulation of the PDF is carried out in the ISSAC program (Sébastien Le Rouxa and Valeri Petkova, *J. Appl. Cryst.* 2010, 43, 181-185), which is an open source program with very informative manual. As for the structural model, we use the AIOC-41 core that is directly obtained from the cif file, and stitched BHET linker to its alcohol ligand site in Material Studio program (the BHET linker can be directly built in the Material Studio program with simple geometric optimization). It is rather obvious that such modeling would only account for the short- and medium-range order within AIOC and the BHET linker, and this is the reason why we use the PDF to analyze the structure of AIOC. As for the overall network connectivity, such information would require extensive fitting of the PDF beyond 1 nanometer, which is beyond the scope of this work and would be studied in the future.

6. The peak intensity in Figure 6a is too low, so it is hard to check the purity and crystallinity of UiO-66@Al-BHET glass ceramic.

We have measured and plotted the data again and shown in Figure S26.

Figure S26. *UiO-66 nanocrystals incorporated in Al-BHET maintain their high crystallinity (a) and surface area (b) as shown by X-ray diffraction and N₂ adsorption. The UiO-66 nanocrystals added to Al-BHET solution also maintains their octahedral morphology after the vitrification of Al-BHET (c).*

7. From the SEM image in Figure 6c, we can see the UiO-66 nanocrystals on the surface of Al-BHET glass. And I am curious about the cross-section image of this glass ceramic.

We have now taken more images that better shows the cracks on the UiO-66@Al-BHET. In fact, as the UiO-66 has large weight percent, the cross-section is not much different from the surface.

Fig. R8. SEM images showing the cross-section of UiO-66@BHET

8. The work about preparing “high-porosity MOF@glass composites” broadens the potential application field of Al-BHET glass, however, this part of content has little to do with the main content of the article. It is better for the author to move this part into the supporting information.

We agree with the reviewer and we have now removed this section and moved to the SI.

Reviewer #5 (Remarks to the Author):

The article by Z. Zhang and Y. Zhao offers an interesting new approach to metal–organic network-forming glasses with good porosity, via a bottom-up synthesis. The article is well written, and I believe that it demonstrates the novelty required to merit publication in Nature Communications, with minor revisions/clarifications.

We are very grateful for the support from the reviewer.

Why does Al-MTBT display a moderate N₂ SA of 362.544 m²/g, yet a lower CO₂ pore volume than Al-BHET-TPTO-2, which displays the lowest N₂ SA?

Compared to N₂, CO₂ molecules have smaller kinetic radius and stronger interactions with the framework, thus can access smaller pore than N₂. For Al-MTBT, it has large pores spanned by the rigid MTBT linker but smaller overall void fraction due to the generally dense framework, thus giving higher N₂ uptake but only moderate CO₂ uptake. Comparably, Al-BHET-TPTO-2 have flexible BHET linker, and the rather large amount of TPTO reduces network connectivity which also adds to the flexibility of the network. It is usually difficult to maintain large pores in such flexible networks, and consequently the Al-BHET-TPTO-2 show low N₂ uptake consistent with small pores. On the other hand, the removal of the rather large amount of TPTO give high overall void fraction accessible to CO₂, thus giving high CO₂ uptake. To avoid the paper being overly lengthy, we now moved the data of Al-BHET-TPTO-2 to the supporting information.

The authors state that “The methanol uptake at room temperature also confirmed the porosity of the material”, though this statement may require a little more explanation. Are these materials stable to water vapour? I wonder if the authors have looked at the water uptake of the materials in a similar way to how methanol uptake was determined?

Fig. R9. Comparison between methanol and water uptake for the aluminum alkoxide glasses and monoliths.

We have measured water vapor uptake in a similar manner as methanol (figs. S32, S46, S16, S6). The AIOCs are generally not thermodynamically stable towards hydrolysis but are stable enough

for a water adsorption measurement. Water and methanol uptake are commonly used to show the porosity of MOF glasses since they can have stronger interaction with the framework than CO₂ and N₂. In our case, methanol and water uptakes are only used as a corroborating evidence for porosity, and the pore volume is calculated from the adsorption data of N₂ and CO₂. As we did not observe a steep uptake of methanol and water at very low pressure but there is also large hysteresis in desorption, we would speculate the methanol and water to have both chemical and physical adsorption to the porous glasses.

The authors note the formation of crystal glass composite materials using the highly porous UiO-66 as the crystalline phase. This approach is similar to several other studies, including *Nat. Commun.*, 2019, 10, 2580 and *Chem. Sci.*, 2020, 11, 9910-9918. I am not sure that this adds much to the study, and perhaps takes away from the interesting properties of the original glasses. Furthermore, the authors use the term “glass ceramics” – terminology with which I disagree. Glass ceramics are produced by selective crystallisation from a base glass, so these composites should be referred to as MOF-CGCs, not glass ceramics. The materials also should not be referred to as hybrid analogues of glass ceramics (*Chem. Eur. J.*, 2022, 28, e202104026), as the crystalline and glass phases are from different materials.

We thank the reviewer for raising this point and we have now change the word “glass ceramics” to “composite” and then moved this section to supporting information.

REVIEWER COMMENTS

Reviewer #1 (Remarks to the Author):

The authors have addressed several concerns raised previously. However, there are some comments that have not been resolved. The details are listed below:

1. In the revised manuscript regarding the terminology used, the author mentioned, "In the activation process, the residue solvent-modulator and unbounded BHET linker are removed from the pore, which makes the pore accessible to CO₂ gases and result in disappearance of T_g in the DSC curve." Does this mean that, after activation, the sample is no longer considered glass? This should be clarified, as well as whether the porosity is the property of the glass or not.

2. I believe that EXAFS data are important for verifying the structure of the proposed glasses in this work. Following my previous comment, the authors have added additional EXAFS data for AIOC-12 as a reference to confirm the local structure of the obtained glass. However, more discussion should be provided, especially regarding why AIOC-12 is a suitable reference. Apart from the mentioned Al–O bond, all other peaks are unmatched. Additionally, the details of AIOC-12 in the discussion are unclear, and the structure of AIOC-12 with EXAFS peak assignments should be provided.

Reviewer #2 (Remarks to the Author):

The authors have addressed the comments and now the paper can be published in its current form.

Reviewer #3 (Remarks to the Author):

The authors have addressed the reviewers' comments and adequately improved manuscript. Therefore, I recommend accepting the manuscript.

Reviewer #4 (Remarks to the Author):

I am very pleased with the authors' excellent and detailed responses to my comments and those of the other reviewers. I appreciate the way they have revised the paper and believe it will be a very useful reference for future work in glass chemistry and beyond.

Reviewer #5 (Remarks to the Author):

The revised manuscript by Zhang and Zhao has addressed the concerns regarding porosity

and terminology which were raised previously, and I believe that the article is now suitable for publication in Nature Communications.

The authors have addressed several concerns raised previously. However, there are some comments that have not been resolved. The details are listed below:

We appreciate the reviewer's dedication to improving this manuscript and clarifying the glassy nature and local structure of the material. We have now further revised our manuscript to address the reviewer's concern and give the audience a very clear picture of these two key matters.

1. In the revised manuscript regarding the terminology used, the author mentioned, "In the activation process, the residue solvent-modulator and unbounded BHET linker are removed from the pore, which makes the pore accessible to CO₂ gases and result in disappearance of T_g in the DSC curve." Does this mean that, after activation, the sample is no longer considered glass? This should be clarified, as well as whether the porosity is the property of the glass or not.

We understand the reviewer's concern and have now added clarification in the manuscript so that the audiences would not have misunderstandings. The porosity and glassy nature of the aluminum alkoxide glasses can be best understood by considering them as host-guest binary systems where the host are the coordinatively linked aluminum alkoxide frameworks and the guests are modulator molecules in the pore. The modulator molecules in the pore provide configurational degree of freedom to the framework and can also be viewed as plasticizers. We believe the aluminum alkoxides should be considered as porous glasses for the following reasons:

(1) **The as-synthesized glasses are porous with the pores occupied by modulators.** In the terminology of material chemistry, "pore" means the voids of robust frameworks where guest molecules can move in and out. Thus, the volume occupied by the modulator in the as-synthesized glass is the pore volume. The activation process, without affecting the coordinatively linked backbone, merely exchanges the modulators in the pore with gases so that the volume occupied by these modulators can be quantified by gas adsorption isotherm. Thus, we should consider the as-synthesized glasses as porous glasses, only that the pores are occupied by modulator molecules.

(2) **The activated networks are glasses because they are the coordinatively linked backbone of the as-synthesized glasses.** The as-synthesized aluminum alkoxide glasses composed of a coordinatively linked network and guest molecules in the pore. As the activation process is a mild physical process that merely remove the guest molecules (modulator) from the pore, the activated glasses are the coordinatively linked backbone of the as-synthesized glasses with well-defined glass transition. Thus, the pores that can adsorb gas in the activated glasses are intrinsic properties of the glassy coordinative backbone. In other words, the activation process removes the plasticizer from a glass so that the glass no longer flows when heated (i.e. T_g disappears), it does not mean the glass without the plasticizer is no longer considered as a glass. This is fundamentally different from calcinating a glass into a porous foam where the pores are generated by out gassing.

From a structural perspective, network-forming glasses are initially described as "random continuous networks" to reflect the fact that glasses are structurally similar to liquids in terms of disordering and randomness (W. H. Zachariasen, *J. Am. Chem. Soc.* **1932**, *54*, 3841). The as-synthesized aluminum alkoxide glasses, with well-defined glass transitions, are kinetically frozen liquids that also have such "random continuous network". As the activation process does not alter

the coordinatively linked backbone of the glass, the “random continuous network” is preserved in the activated glasses even though the empty framework does not have a glass transition. In other words, the structures of the activated glasses represent the continuous subsets of glassy materials (i.e. the as synthesized glasses) and should also be considered as glasses.

(3) From a practical perspective, the motivation to develop metal-organic network-forming glasses is to have materials that combines the processability of glasses and the porosity of MOFs. The aluminum alkoxide glasses can achieve this as they can be first shaped in the as-synthesized form before in situ activation to give porosity.

However, we surely acknowledge that, the aluminum alkoxide glasses are fundamentally different from glasses that show porosity immediately after quenching from the super-cooled liquid (such as melt-quenched glasses of ZIF-62). It remains a fundamental challenge to design network-forming glasses that have high porosity immediately after quenching from the super-cooled liquid state (glass transition), which we would also wish to pursue in the future.

In addition, from a historical perspective, the term “porous glasses” could refer to etched phase-separated glasses, which also have no porosity right after quenching from liquid states (Porous glasses in the 21st century—a short review. *Microporous and Mesoporous Materials* **2003**, 60, 19). The reason why they are called “glasses” is that they can be first shaped as regular glasses and then etched to give large mesopores, and the etched glasses also retain the continuous networks of glasses quenched from super-cooled liquids. In this perspective, the aluminum alkoxides networks should certainly be categorized as porous glasses, only that they show micropores after physically exchanging out the modulators whereas the traditional inorganic porous glasses show mesopores after chemical etching.

The essence of the discussion above is added to the last paragraph of the introduction and discussion

“Comparison between the glass transition behavior, compositions and structure features of the as-synthesized and activated glasses shows that modulator molecules can have both pore-templating and plasticizing effect for the coordinatively linked network, which enable the combination of processability and porosity for the aluminum alkoxide glasses in a unique way: the as-synthesized glasses, with modulators in the pore, show well-defined glass transitions and rheological behavior and can be processed like regular glasses; the activated glasses, with the modulator removed from the pore, show high gas uptake but absence of glass transition, and thus have fixed morphology and would not flow”

“Thus, the activated glasses, although without well-defined glass transition, are structurally identical to the as-synthesized glasses in terms of the coordinatively linked backbone, and the only difference between them is whether the pore is filled with modulator molecules. Consequently, the activated glasses also inherit the transparency and homogeneity of the as-synthesized glasses. From the perspective of the as-synthesized glasses, the activation and gas adsorption process merely exchange the modulator in the pore with gases. As the term “pore” means the voids of a robust framework where guest molecules can move in and out, the as-synthesized glass should also be considered as porous. However, the porosity of the aluminum alkoxide glasses is fundamentally different from the porosity of zeolitic imidazolate framework glasses, which is open to guest

molecules immediately after quenching from the super-cooled liquid without the need for any additional activation process. From a practical perspective, the aluminum alkoxide glasses can be first shaped in the as-synthesized form and then activated, thus combining the processability of glasses, porosity of crystalline frameworks and the modular designability of reticular chemistry.”

2. I believe that EXAFS data are important for verifying the structure of the proposed glasses in this work. Following my previous comment, the authors have added additional EXAFS data for AIOC-12 as a reference to confirm the local structure of the obtained glass. However, more discussion should be provided, especially regarding why AIOC-12 is a suitable reference. Apart from the mentioned Al–O bond, all other peaks are unmatched. Additionally, the details of AIOC-12 in the discussion are unclear, and the structure of AIOC-12 with EXAFS peak assignments should be provided.

The local structure of the Al-BHET is primarily studied by PDF and the EXAFS is only used as corroborating evidence to support the presence of AIOCs. This is because unlike transition metals that can be routinely studied by EXAFS to determine their coordination environment, the EXAFS of aluminum is difficult to measure and analyze, especially for non-crystalline materials (ref. 23, see also *J. Phys. Chem. C* **2007**, 111, 11721). The difficulty in measuring Al EXAFS is associated with its K-edge position, which is in the soft X-ray regime. In addition, the Al-BHET is highly insulating, further reducing the signal-to-noise ratio.

Fig. R1. The k-space plot for the EXAFS data for Al-BHET and AIOC-12

From the k-space plot of the EXAFS data, it can be found that the noise level is already high at 5 Å⁻¹, and for quantitative fitting for the first shell or qualitative analysis of the second shell, the k-space data would need to be smooth at least to 10 Å⁻¹. Consequently, the peaks in the EXAFS R-space plot other than the first one corresponding to the Al–O bond length are dominated by noise and carry no reliable information, so these peaks cannot be assigned. Thus, the only reliable information we can get from X-ray absorption spectroscopy is the near edge feature that correlates the valency of aluminum and the qualitative information of the first shell (i.e. the average Al–O length). As we have explained before, comparing the near edge features of alumina, AIOC-12 and Al-BHET provide an important corroborating evidence that indicates the chemical environment of aluminum in Al-BHET is consistent with AIOCs and different from alumina. The reason for choosing AIOC-12 as a reference is because AIOC-12 is a stable and synthetically viable AIOC that is similar to AIOC-41 in terms of Al coordination environment (mostly octahedral), type of ligands (mixed bridging OH, carboxylates, alkoxides) and Al–O bond length (presence of diverse Al–O bond lengths that are mostly above 1.85 Å. These features can be clearly seen from the figures below that compare different AIOCs and frameworks.

Fig. R2. The structure and Al-O bond lengths of aluminum alkoxides with pure alcohol ligands (Ref: *Polyhedron* **2022**, 223, 115958), the aluminum are largely in tetrahedral coordination environments with relatively short bond length (below 1.85 Å).

Fig. R3. The structure and Al-O bond lengths of AIOC-12 (ref 17). AIOC-12 has mixed carboxylic acid and alcohol ligands, and the aluminum are all in octahedral coordination geometry with relatively long and diverse bond length (above 1.85 Å).

Fig. R4. The structure and Al-O bond lengths of AIOC-41 (ref 20). AIOC-41 has mixed carboxylic acid and alcohol ligands, and the aluminum atoms are mostly in octahedral coordination geometry with relatively long and diverse bond length (above 1.85 Å).

Fig. R5. The structure of MIL-53 with all carboxylic acid linkers. The aluminum atoms are in simple octahedral coordination geometry with only three types of Al-O bonds. Ref: *J. Am. Chem. Soc.* **2008**, 130, 14170.

REVIEWERS' COMMENTS

Reviewer #1 (Remarks to the Author):

The authors addressed all my concerns.

REVIEWERS' COMMENTS

Reviewer #1 (Remarks to the Author):

The authors addressed all my concerns.

We thank the reviewers for their support